# Boron Isotopes in Fresh Surface Waters in a Temperate Coastal Setting

**Brooke N. Peritore [1], E. Troy Rasbury [1],\*[ID], Kathleen M. Wooton [1], Carrie C. Wright [1][ID], Deanna M. Downs [1], Anastasia Iorga [2][ID] and Shannon L. Letscher [1]**

1   Department of Geosciences, Stony Brook University, Stony Brook, NY 11794-2100, USA;
    brooke.peritore@stonybrook.edu (B.N.P.); katie.wooton@stonybrook.edu (K.M.W.);
    carrie.wright@stonybrook.edu (C.C.W.); deanna.downs@stonybrook.edu (D.M.D.);
    shletsch@gmail.com (S.L.L.)
2   Interdepartmental Doctoral Program in Anthropological Sciences, Stony Brook University,
    Stony Brook, NY 11794-2100, USA; anastasia.iorga@stonybrook.edu
\*   Correspondence: troy.rasbury@stonybrook.edu

**Abstract:** The results from a four-year study of a freshwater pond on Long Island, NY, USA, do not point to a single source of boron (and by proxy other elements including nutrients) in this system. However, boron data from samples associated with this pond can be explained by mixing between average precipitation (weighted average $\delta^{11}$B = 22.7) in the area and the local sources of boron, both natural and anthropogenic. This multiyear study provided the opportunity to see both yearly and seasonal differences. One algae sample from the pond showed significant fractionation and enrichment in light boron relative to the water and suggests algae may act as a boron sink. This type of biological fractionation could explain an observed down-gradient trend to heavier boron isotope values in pond water, which corresponds to the slight reduction in boron concentration seen in 2021. However, the trend was subdued in the following year, likely due to differences in the water flow rates and/or rate of algal growth. An opposite trend was seen with depth in the water, where $\delta^{11}$B showed a positive correlation to boron concentration, which increased with depth from the surface of the pond. This gradient may be explained by the stratification of the pond with a heavy source concentrating in the bottom waters. The bottom water composition was consistent with goose feces ($\delta^{11}$B = 25.8) or the addition of chemicals from the application of rock salt to local roads in winter. Surprisingly, boron from seawater (average $\delta^{11}$B = 39.8) did not appear to have a direct impact on Setauket Pond, other than its influence on precipitation, providing heavy $\delta^{11}$B and very low boron concentrations.

**Keywords:** fertilizer; septic; waterfowl; hydrology; algae; $\delta^{11}$B

## 1. Introduction

Boron is a conservative element with a wide range of natural isotope ratio values in the Earth's critical zone [1], making it a useful element for environmental research. Numerous studies used boron isotopes as a proxy tracer of environmental contaminants [1–8]. This type of study is possible because boron (B) is added to common environmental contaminants, such as fertilizers and detergents [5]. Boron may also enter the environment from the leaching of landfill materials [9]. Natural additions of boron also occur through precipitation (rain or snow) events and interactions between water and soil [10].

Boron has two stable isotopes, namely, $^{11}$B and $^{10}$B, that occur in the aqueous environment in the forms of trigonal boric acid B(OH)$_3$ and tetrahedral borate ion B(OH)$_4^-$ [11]. The proportion of boric acid to borate is dependent on the pH, with boric acid dominant at lower pH (most groundwaters) and borate dominant at higher pH. A pH-driven measurable fractionation of B isotopes between boric acid and borate in solution [12] and differences in the chemical behavior of boric acid and borate may result in changes in the

B isotope composition of waters. For example, all known sources of B entering the ocean, such as rivers [13,14], are isotopically lighter than seawater. Coprecipitation or sorption onto minerals favors borate (depleted in $^{11}$B), which leaves seawater isotopically B heavy (enriched in $^{11}$B) [13,14].

Despite the many studies utilizing boron, a regional background for sources is still needed to take advantage of the ability boron isotopes have to act as a contaminant tracer. For example, proximity to the ocean with its elevated concentrations and heavy isotope ratio may impose a signature in coastal settings. This study specifically sought to establish better constraints for using boron in temperate coastal settings by using the Setauket Pond on the northern shore of Long Island, NY, USA, as a natural laboratory. Long Island was the first location in the United States to be designated as a sole source aquifer. Surface waters recharge this aquifer, making it important to understand the source and pathway of contaminants.

An earlier study of boron that focused on sources of nitrate into the Long Island Sound via a subterranean groundwater discharge (SGD) found that $\delta^{11}$B combined with $\delta^{15}$N and $\delta^{18}$O of nitrates in water samples do not provide values that fit cleanly within the range of published values for its use as a tracer of nitrates [15]. However, while this study could not pinpoint specific sources, it did identify differences between the two study areas. These two areas, which were selected to represent agricultural (fertilizers and manure) vs. urban sources (septic and fertilizer), both showed a large range in $\delta^{11}$B [15]. The agricultural site had higher boron concentrations and lighter $\delta^{11}$B, while the urban source area had lower boron concentrations that trend toward the seawater $\delta^{11}$B value [15]. This study revealed the potential to use boron isotopes to identify sources, but a better framework for deciphering these sources needs to be established for this coastal setting. To this end, our research focused on a small spring-fed freshwater pond (Setauket Pond) that is near Stony Brook University, providing the opportunity to include undergraduate students in the research and to sample over multiple years.

The focus of the research was to investigate the different sources and possible sinks of boron for Setauket Pond by comparing a variety of waters (precipitation, pond, and culvert) and known natural (aquatic wild bird feces, algae) and anthropogenic boron sources and sinks (fertilizer, septic, and manure), and to consider changes to boron concentration [B] and isotope values ($\delta$11B) due to processes in this anthropogenically influenced spring-fed pond.

## 2. Study Area

### 2.1. Background

Long Island is in the southern part of New York on the east coast of the United States (Figure 1). NOAA precipitation and temperature records for the region (Islip MacArthur Airport, NY, weather station) show that from 1991 to 2020, there was an average of 46.1 inches/1170.9 mm of precipitation per year, with precipitation spread evenly across seasons (winter: 11.7 inches/297.2 mm, spring: 11.9 inches/302.3 mm, summer: 11.5 inches/292.1 mm, and autumn: 11.0 inches/279.4 mm) [16]. The average temperature from 1991 to 2020 for the region was 53.1 °F/11.7 °C, though there are well-developed seasons, with winter temperatures averaging around the freezing point of water (34.1 °F/1.2 °C) and summer temperatures averaging (T72.6 °F/22.6 °C). Winter storms impacting Long Island are generally northeaster ('nor'easters' develop within 100 miles of the East Coast between Georgia and New Jersey before moving northward along the coast while dropping heavy rain and/or snow, especially over New England and the eastern coastal provinces of Canada) [17], while most summer rains result from local moisture sources, with hurricanes bringing precipitation from far away sources.

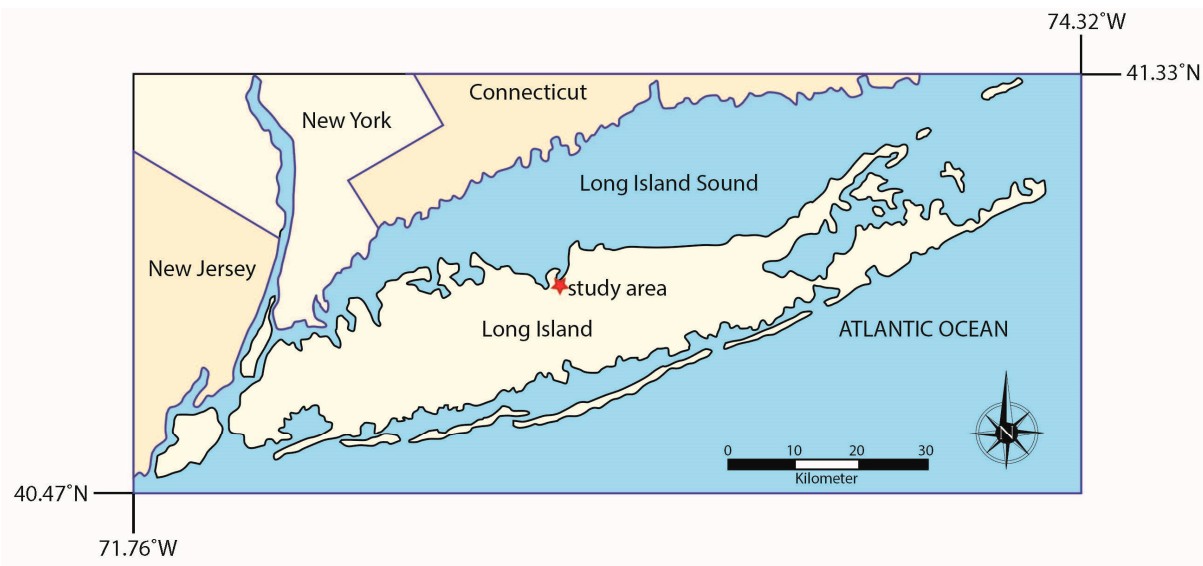

**Figure 1.** Map of Long Island showing the study area on the northern shore.

Long Island is underlain by a crystalline basement that is a continuation of the basement rocks exposed to the north of the Long Island Sound (LIS) in Connecticut, USA. Cretaceous units of the Lloyd, Raritan, and Magothy Formations unconformably overlie this basement. The basement and Cretaceous strata dip to the south such that the Cretaceous is exposed in several places along the northern shore of Long Island. Pleistocene-age glacial deposits, including the Jamico Aquifer and the Gardiners Clay, overlie the Cretaceous strata (Figure 2).

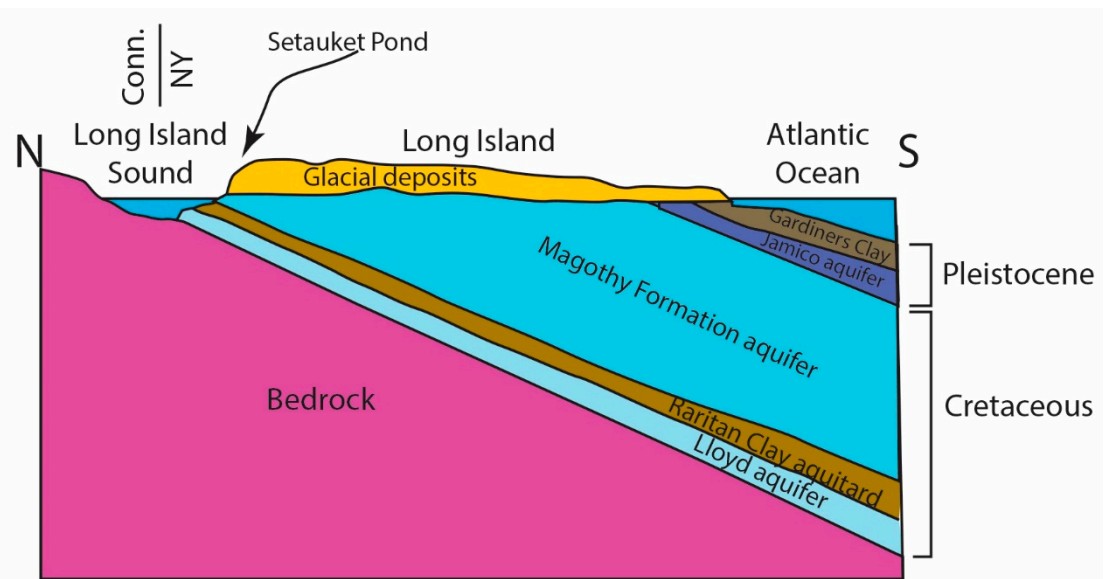

**Figure 2.** Cartoon cross-section of Long Island geology from north to south showing the aquifers and aquitards. The topography of two glacial moraines controls drainage divides. This cartoon is vertically exaggerated and not to scale and is based on work from Cohen et al. [18].

### 2.2. Upper Glacial Aquifer

The Upper Glacial Aquifer is the uppermost aquifer on Long Island and the water table occurs within this unit (Figure 3). Precipitation infiltrating the Upper Glacial Aquifer recharges the Magothy Aquifer (Figure 3), which provides an estimated 90% of the drinking water to Nassau County (west) and 50% of the drinking water to Suffolk County (east) on Long Island. The Magothy Aquifer mostly consists of coarse to fine sand, resulting

in moderate permeability, with localized areas of higher or lower permeability with the presence of gravel or silt and clay [18,19]. Glacial deposits are known for their heterogeneity, where the regional flow is separated into non-interacting, quasi-parallel systems, which result in spring-fed creeks that flow generally from south to north on the northern side of the glacial moraine (Figure 3). Permeability differences in the overlying glacial moraine sediments result in "compartmentalization" such that the springs and ponds nearby the Setauket Pond have distinctive and different boron isotope values, as shown in Tamborski (Table 1) [1] and with additional examples in our Table 1.

**Table 1.** Boron concentration and isotope ratio values from Long Island Ponds in 2019.

| Sample Name | $\delta^{11}B$ (‰) | 2SD | B (ppb) | Latitude | Longitude |
|---|---|---|---|---|---|
| Blydenburg Pond A, Smithtown | 10.4 | 1.2 | 48 | 40.8428618 | −73.2277239 |
| Blydenburg Pond B, Smithtown | 17.9 | 0.9 | 40 | 40.8429112 | −73.2278610 |
| Blydenburg Pond C, Smithtown | 12.8 | 0.6 | 39 | 40.8419654 | −73.2281765 |
| Blydenburg Pond D, Smithtown | 14.1 | 0.0 | 53 | 40.8381946 | −73.2298402 |
| Blydenburg Pond E, Smithtown | 12.9 | 0.5 | 45 | 40.8364809 | −73.2259872 |
| Grassy Pond, Calverton | 31.5 | 0.1 | 9 | 40.8932991 | −72.8371785 |
| Sandy Pond, Calverton | 32.3 | 1.5 | 5 | 40.8959775 | −72.8374258 |
| T. Bayles Minuse Mill Pond A, Stony Brook | 15.8 | 0.4 | 24 | 40.913765 | −73.146406 |
| T. Bayles Minuse Mill Pond B, Stony Brook | 14.2 | 0.3 | 25 | 40.913652 | −73.147568 |

2SD is two-sigma standard deviation of the mean.

Precipitation from near Stony Brook University was collected over the span of this study. Some of it is published [15,20] and new data are also presented in this contribution (Table 2). The wide range in the $\delta^{11}B$ of precipitation sets the stage for the wide range of values we found in the groundwater. However, elevated [B] relative to the precipitation requires additional sources to the groundwater system to achieve the concentrations observed.

**Table 2.** Precipitation data.

| Sample Type | Date | Wind Direction | Average Temperature (°C) | $\delta^{11}B$ (‰) | 2SD | B (ppb) |
|---|---|---|---|---|---|---|
| Rainwater | 6/9//19 | NE | 17 | 15.6 | 0.3 | 7 |
| | 27/10/20 | SE | 16 | 33.4 | 0.3 | 6 |
| | 12/11/19 * | NW | 5 | 21.3 | 0.3 | 9 |
| | 29/10/20 | E/NE | 10 | 31.5 | 0.4 | 4 |
| | 30/10/20 | N/NE | 5 | 34.0 | 1.0 | 4 |
| | 1/11/20 | S | 10 | 4.7 | 0.4 | 5 |
| | 23/11/20 + | SW | 12 | 26.5 | 0.4 | 5 |
| | 4/12/20 | W/SW | 9 | ˆ | - | ˆ |
| | 14/12/20 | N | 3 | 13.9 | 0.4 | 2 |
| | 1/1/21 | E/NE | 2 | 27.6 | 0.4 | 1 |
| | 16/1/21 | N/mixed | 7 | 30.8 | 0.4 | 5 |
| | 26/1/21 * | E/NE | 1 | 25.3 | 0.4 | 5 |
| | 16/2/21 | S/mixed | 6 | 12.3 | 0.4 | 5 |

**Table 2.** *Cont.*

| Sample Type | Date | Wind Direction | Average Temperature (°C) | $\delta^{11}$B (‰) | 2SD | B (ppb) |
|---|---|---|---|---|---|---|
| Snow | 15/12/20 | NE | −2 | 21.8 | 0.4 | 1 |
| | 20/12/20 | Calm/N | 0 | 10.4 | 0.4 | 1 |
| | 2/1/21 | NE | −1 | 27.0 | 0.4 | 1 |
| | 1/2/21 | NE | −1 | 19.0 | 0.4 | 1 |
| | 2/2/21 | N | 1 | 30.4 | 0.4 | 1 |
| | 7/2/21 | N/NE | 0 | ˆ | - | ˆ |
| | 11/2/21 | N/NE | −3 | 25.1 | 0.4 | 1 |

2SD is two-sigma standard deviation of the mean. * Rain/sleet, + thunderstorm, and ˆ larger sample quantity is needed for results.

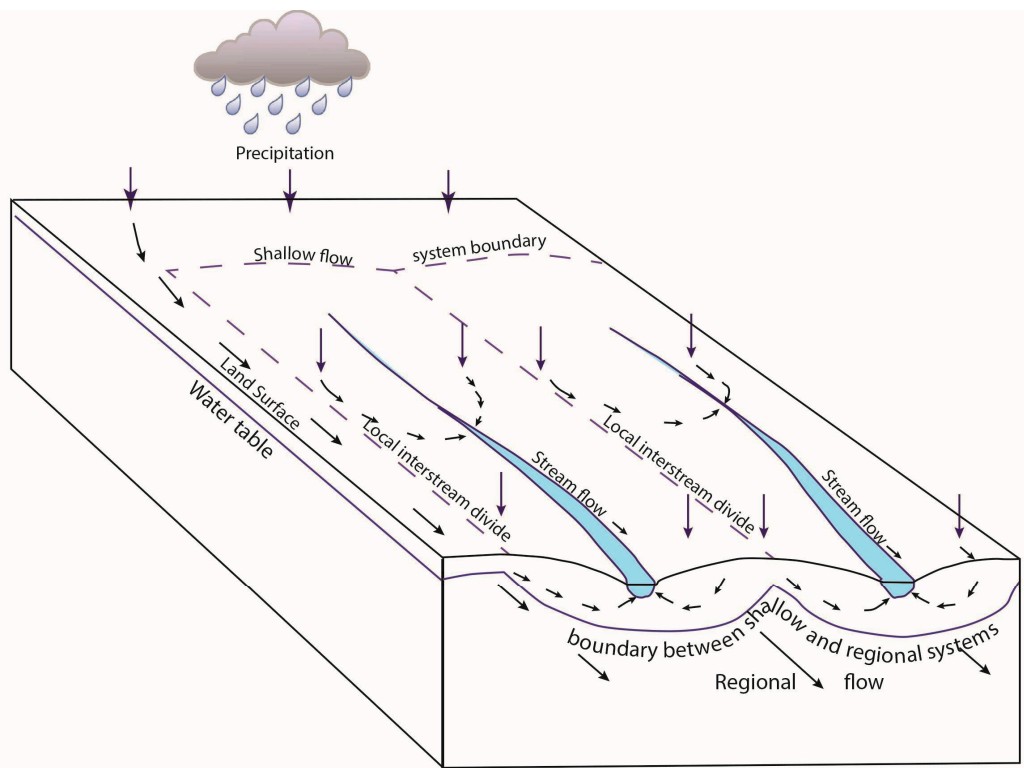

**Figure 3.** Schematic diagram showing the relationship between the Upper Glacial Aquifer and the regional aquifer system from Prince et al. [19]. Setauket Pond can be represented as one of the streams with local spring input. All of these streams flow toward the north on the northern side of the moraine and toward the south on the southern side of the moraine. The system is "compartmentalized" because these closely spaced streams have different boron isotope compositions from each other, which would not be the case if there was significant mixing. The arrows represent the direction of water flow from precipitation through the system.

Setauket Pond Natural Laboratory

The Setauket Pond occupies a low-elevation area that is separated from Conscience Bay of the LIS by a dam. It is one of many man-made, spring-fed ponds on the northern shore of Long Island that have a water table near the surface [21]. The spring-sourced water originates near Detmer Farm and is transported via Setauket Creek to Setauket Pond (Figure 4). On its journey from the spring to the pond, Setauket Creek flows from near Detmer Farm and through medium-density housing communities, which are all potential sources of boron via lawn/garden treatments and septic systems (though lots in Setauket tend to be heavily wooded). The Setauket Pond has southern and northern parts that

are connected under the Old Field Road stone arch bridge (Figure 4), which acts as a constriction point between the two sections of the pond. The northern pond is separated from the LIS by a dam at the site of an old grist mill in the Frank Melville Memorial Park. The land around the northern pond is part of this park with uninhabited historic buildings in the park (old barn, old mill house, etc.). There are few residences near this part of the pond that can contribute anthropogenic boron to the system (Figure 4). This is unlike the southern pond, where there is medium-density housing, the Three Village Tennis Club, and the historic Setauket Neighborhood House (just south of the Three Village Tennis Club), all of which are possible sources of boron from lawn and garden treatments and runoff, as well as through their septic systems (Figure 4). Setauket Elementary School and Gelinas Middle School, with their large fields, are additional potential anthropogenic sources of boron to Setauket Pond.

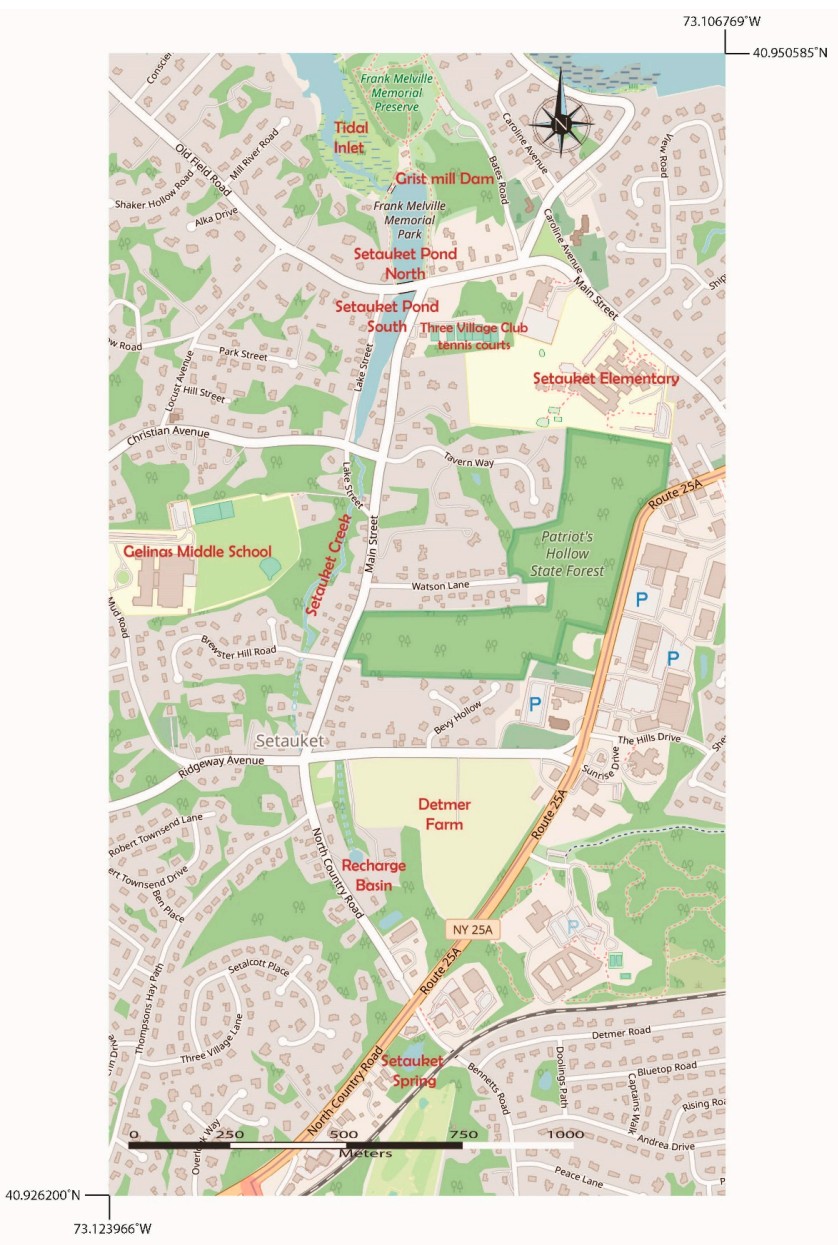

**Figure 4.** Map showing the northern and southern portions of Setauket Pond, the location of the Three Village Tennis Club, Gelinas Junior High School, Setauket Elementary School, and Frank Melville Memorial Park, which consists of native woodland and a Long Island Sound estuary/tidal inlet environment. West of Detmer Farm is Setauket Creek flows from a freshwater spring originating south of Detmer Farm and feeds into the southern portion of Setauket Pond.

The water flow velocities and water levels of creeks that combine to deliver water to the Setauket Pond change throughout the year in response to rainfall amounts, and are typically lowest in the late summer and early fall. Based on this observation, source springs appear to be responsive to rainfall. Seasonally the pond experiences algal blooms, and some of these are harmful, based on warning signs posted next to the pond. By the start of fall, the pond reaches maximum green algae coverage, with algae die-off occurring over the winter, removing green algae from the system. By early summer (July), algae have noticeably returned to the pond and can be found in a progressive gradient from the highest concentration of the plants in the southern portion of the southern pond to the lowest in the northern portion of the northern pond. This trend will be maintained, even as the algae overall will increase in the pond over the summer. Algae are not the only plants in the pond and future work should focus on the identification of the plant biomass.

Because the spring-fed creek and Setauket Pond are a constrained system with identifiable sources of boron that are also in close proximity to Stony Brook University, this study location offers both easy access for repetitive monitoring and sampling, as well as being optimally positioned to study groundwater where it intersects the surface. These features make the Setauket Pond an excellent natural laboratory for studying boron systematics in a temperate coastal region.

## 3. Methods

### 3.1. Sampling

We analyzed a total of 91 samples from the Setauket Pond over the past four years as part of programs to refine our understanding of groundwater on Long Island, as well as to include undergraduate students in research [20,22–24]. To characterize possible anthropogenic inputs, we also collected a variety of potential fertilizers and septic system waters. Additionally, we collected rain and snow samples to build on published values reported in Tamborski et al. [15].

All water samples were collected in clean 125 mL Nalgene narrow-mouth bottles. Water samples were collected from near the shores of Setauket Pond and from the center of the pond using a kayak. Water samples in some near-shore locations in the southern portion of the pond also included surface and below-surface sampling. This was accomplished by wading near the shore and, for each sample location, water was collected at the surface, at approximately the halfway point between surface and bottom, and near the bottom. This was done to determine any trends in the [B] and $\delta^{11}B$ values with depth. Water from two culverts adjacent to the pond where runoff was focused and captured was also sampled. Repeated collection from four sites was conducted over four years to investigate seasonal and yearly variability of [B] and $\delta^{11}B$ values.

Precipitation, rainwater, and snow were collected in 5 L pre-cleaned Tupperware containers. Goose and swan feces were collected using nitrile gloves and placed into Ziplock bags. Fertilizers, including manure, were sampled from locally purchased sources that might have been used on lawns and in gardens near the pond. These were put in clean 50 mL centrifuge tubes. Septic system samples from nearby residences were provided by the New York Center for Clean Water Technology.

Supporting data collected included GPS coordinates, temperature, and nitrate concentrations. The temperature was measured with a thermometer that was bundled with a Hanna Instruments pHep. The pH measurements taken with this instrument were calibrated with Milwaukee calibration buffer solutions. Nitrate concentrations were measured with a Vernier Nitrate Ion-Selective Electrode calibrated with sodium nitrate standard solution. Unfortunately, due to the probe overheating and only being available in 2022, only a few nitrate analyses are reported.

The solid samples were leached to obtain boron. The Har Tru clay sample was leached in 2 M nitric for 1 week. Fertilizers were leached in Milli-Q water for a week. After the one-week leach, aliquots of the supernatants were taken to process for boron isotopes. The

supernatants and water samples were filtered through nylon 0.22 mm filters to remove organic materials before ion exchange chemistry was performed.

In preparation for the B purification chemistry, all filtered samples were adjusted to a pH of ~9 using high-purity ammonia (Baseline™). Samples were loaded on the boron-specific resin Amberlite IRA 743 in a method modified from Hemming and Hanson [25] and Lemarchand et al. [13] using a Watson-Marlow 16-channel peristaltic pump with 1 mL pipette tips fitted with a cellulose frit to hold the resin in. Seawater was put through B purification chemistry alongside each batch of samples as a test of the reliability of the chemistry, including the completeness of B recovery and boron isotope values. Ions other than boron were washed from the columns with pH-9-adjusted Milli-Q water. Boron was eluted with 1.2 mL of the same 2% nitric acid used during the mass spectrometry analysis.

### 3.2. Analysis

A Nu Instruments Plasma II multicollector inductively coupled plasma mass spectrometer (MC-ICP-MS) was used for B isotope analysis. Small aliquots from each eluant were diluted to 10% and tested for concentration via comparison with a 50 ppb standard. Based on this, samples were diluted to signal match 50 ppb of the standard NBS 951. A common batch of 2% nitric acid was used during the B purification chemistry, for diluting samples, and during instrumental analysis. This was done to minimize any matrix bias and to normalize the backgrounds.

Boron was run as a wet plasma using a Glass Expansion "twizzler" type spray chamber that was cooled to 7 °C with a Peltier system. A 100 mL/min Glass Expansion quartz glass nebulizer was used to aspirate the solution into the spray chamber. The sample was injected into the plasma through a Glass Expansion glass torch (31-808-3087). Nu Instruments "wet" light isotope sampler (WA1.15) and skimmer nickel cones (WA7) were used for introduction into the mass spectrometer. We measured $^{11}B$ in Faraday cup H5 and $^{10}B$ in Faraday cup L6 and aligned these masses by adjusting the quad values. There was a small peak due to quadruply charged Ar on the left shoulder of the $^{10}B$ peak; however, it was far enough from the peak center that it did not bias our results (but was always monitored). During the analysis, the standards and samples were bracketed using the same common 2% nitric acid batch, which allowed for the subtraction of the small (<1 ppb) addition of boron by the acid. For boron isotopes, unknowns were referenced to the boric acid standard NBS 951 and $\delta^{11}B$ designates the value in per mil relative to this standard. The equation used to calculate boron isotope values is

$$\delta^{11}B = \frac{\left(^{11}B/^{10}B_{Sample} - \, ^{11}B/^{10}B_{951}\right)}{\left(^{11}B/^{10}B_{951}\right)} \times 1000$$

Repeat analyses of seawater from Smith Point, New York (Atlantic Ocean of Long Island), during this study gave an average $\delta^{11}B$ of 39.8 ± 0.3‰ (n = 20; 2σ). This is well within the uncertainty of the accepted value of 39.61 ± 0.04‰ [26]. When possible, each sample was run in triplicate to ascertain the reproducibility, and the error is reported as 2 SDs for these replicate runs. For samples with concentrations that were too low to allow for multiple runs, we used a conservative estimate of ±1‰.

## 4. Results

### 4.1. Precipitation

Precipitation samples were collected over the span of the study, though not every event was sampled (Table 2). Snow samples tended to have lower [B] than rainwater. Winter storms for Long Island typically form along the Atlantic coastline between Georgia and New Jersey before moving north and impacting the North Atlantic States and Canadian Provinces (e.g., nor'easters), while summer storms tend to develop along regional cold fronts (e.g., air mass thunderstorms) with occasional remote source precipitation via tropical

storms/cyclones [27,28]. While the [B] values were different, the ranges of $\delta^{11}$B values were similar between the rain and snow.

The snow samples had a very low [B], with ranges from too low to measure to 1.4 ppb and $\delta^{11}$B values of 10.4 to 30.4‰, with a weighted mean of 1 ppb and $\delta^{11}$B of 22.3‰. One snow sample had a $\delta^{11}$B of 10.4‰, which was anomalous relative to the rest, which were mostly > 20‰ (Table 2). Leaving it out, the weighted mean $\delta^{11}$B was 24.7‰. The rain samples had a highly variable range of [B] from too low to measure to 9 ppb (Table 2). The $\delta^{11}$B values for the rain ranged from 4.7 to 33.4‰. The average [B] for all the precipitation events, including snow and rain, was 4.8 ppb and the average $\delta^{11}$B was 22.7‰. However, we do not know the relative amounts of precipitation that fell during these events, and we did not sample every event. These precipitation values thus provide a general baseline and range for the possible input of boron via precipitation.

### 4.2. Setauket Pond and Peripheral Water Samples

To better understand the boron systematics of the surface waters at Setauket Pond, water samples were collected from peripheral locations with the potential to influence [B] and $\delta^{11}$B values (Table 3). Setauket Pond is on the edge of a LIS tidal inlet and is separated from this marshland by the Grist Mill Dam ([B] = 2027 ppb, $\delta^{11}$B = 38.9‰). This allows Setauket Pond water to spill over the top of the dam and into the tidal creek marshland below. The dam also prevents tidal creek water from entering the pond.

Nearby Detmer Farm is of interest from both the perspective of contributing boron into Setauket Pond via runoff into Setauket Creek and because there is a nearby recharge basin that we call Detmer Pond ([B] = 14 ppb, $\delta^{11}$B = 9.8‰), which is just west of the farm's fields (Figure 4). The pond will go dry during droughts, and during periods of heavy precipitation, will spill over on its northern side, creating a temporary creek ([B] = 14 ppb, $\delta^{11}$B = 12.1‰). Setauket Creek is fed from a spring south of Detmer Farm. Two samples of the creek were collected to see a creek value just before entering the pond, Setauket Creek Confluence 1 ([B] = 27 ppb, $\delta^{11}$B = 10.1‰), and where the pond and creek meet, Setauket Creek Confluence 2 ([B] = 44 ppb, $\delta^{11}$B = 10.6‰) (Table 4).

**Table 3.** Peripheral water samples collected from near Setauket Pond in 2019.

| Sample Name | $\delta^{11}$B (‰) | 2SD | B (ppb) | Latitude | Longitude |
|---|---|---|---|---|---|
| Detmer Farm Pond | 9.8 | 0.8 | 14 | 40.933050 | −73.117200 |
| Detmer Farm Creek | 12.1 | 1.0 | 14 | 40.934100 | −73.117700 |
| Tidal Creek | 38.9 | 0.9 | 2027 | 40.947060 | −73.116410 |

2SD is two-sigma standard deviation of the mean.

### 4.3. Multiyear Sampling of Setauket Pond

To help establish the normal range of [B] and $\delta^{11}$B values from one year to the next, five Setauket Pond locations were selected for sampling on an annual basis (Table 5, Figures 5 and 6), though in one instance, the multiyear sampling was accidental. SP7 and SP14 were collected from the same location in different years, 2019 and 2021, respectively. To simplify the discussion, the sample name for this location has become SP7. Three of the multiyear locations are in the northern portion of Setauket Pond and had mean values of SP2 [B] = 30.5 ppb, $\delta^{11}$B = 12.7‰; SP3 [B] = 23.7 ppb, $\delta^{11}$B = 12.4‰; and SP5 [B] = 25.7 ppb, $\delta^{11}$B = 12.2‰, while two were in the southern portion of the Setauket Pond and had mean values of SP1 [B] = 71 ppb, $\delta^{11}$B = −8.4‰ and SP7 [B] = 26.5 ppb, $\delta^{11}$B = 10.7‰. While there were small differences from one year to the next, the mean range for the locations was narrow, namely, [B] = 20 to 30.5 and $\delta^{11}$B = 10.8 to 12.7‰, with the exception being SP1 with a relatively higher average [B] of 71 ppb and an unusual negative $\delta^{11}$B value of −8.4‰. SP4, while not a location for multiyear sampling, is included here because it was in the early collection and provides a baseline for comparison.

**Table 4.** Setauket Pond (south) near-shore western side samples from 2021.

| Sample Name | $\delta^{11}B$ (‰) | 2SD | B (ppb) | Latitude | Longitude |
|---|---|---|---|---|---|
| Setauket Creek Confluence 1 | 10.1 | 0.3 | 27 | 40.941534 | −73.117105 |
| Setauket Creek Confluence 2 | 10.6 | 0.3 | 44 | 40.941700 | −73.117000 |
| Big Field | 5.5 | 0.4 | 32 | 40.942420 | −73.116947 |
| H26 | 11.8 | 0.0 | 26 | 40.942841 | −73.116836 |
| H28 | 13.2 | 0.2 | 24 | 40.943210 | −73.116679 |
| H32 | 9.1 | 0.1 | 27 | 40.943453 | −73.116541 |
| H34A | 11.6 | 0.3 | 33 | 40.943624 | −73.116486 |
| H34B | 14.2 | 0.1 | 24 | 40.943910 | −73.116390 |
| H36 | 17.1 | 0.1 | 31 | 40.944179 | −73.116302 |
| H38 | 11.8 | 0.5 | 42 | 40.944422 | −73.116226 |

2SD is two-sigma standard deviation of the mean.

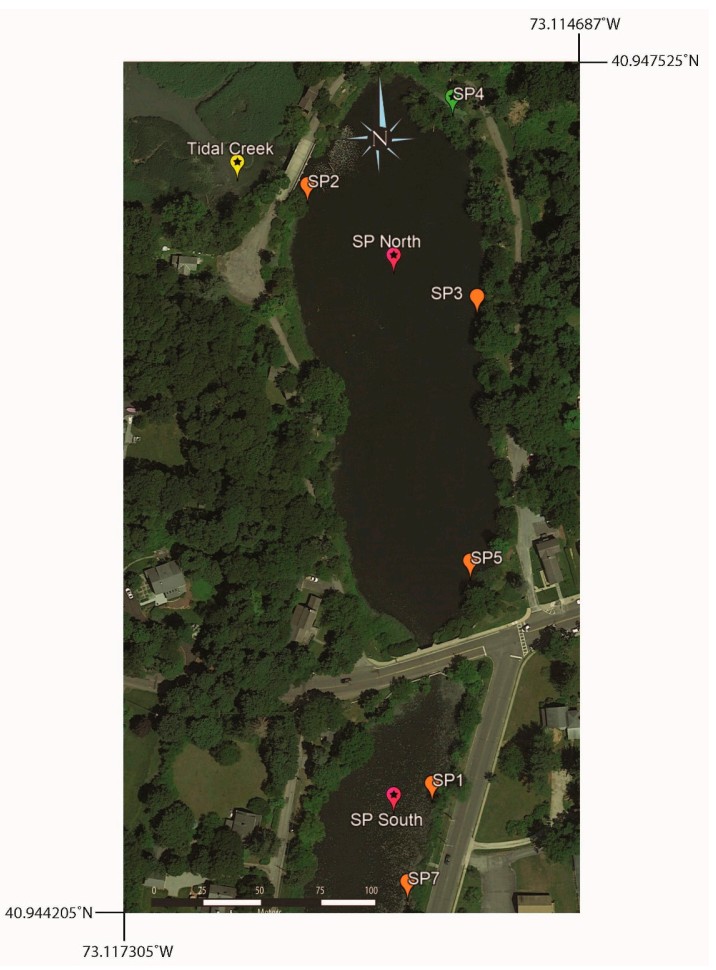

**Figure 5.** Map of multiyear surface water sampling locations: SP1, SP2, SP3, SP5, and SP7. SP4 was collected in 2019 and served as a baseline for the northern end of the lake, even though it was only sampled once. Map also shows the location of the Tidal Creek and the walkway over the top of the Grist Mill Dam, which is adjacent to sample location SP2.

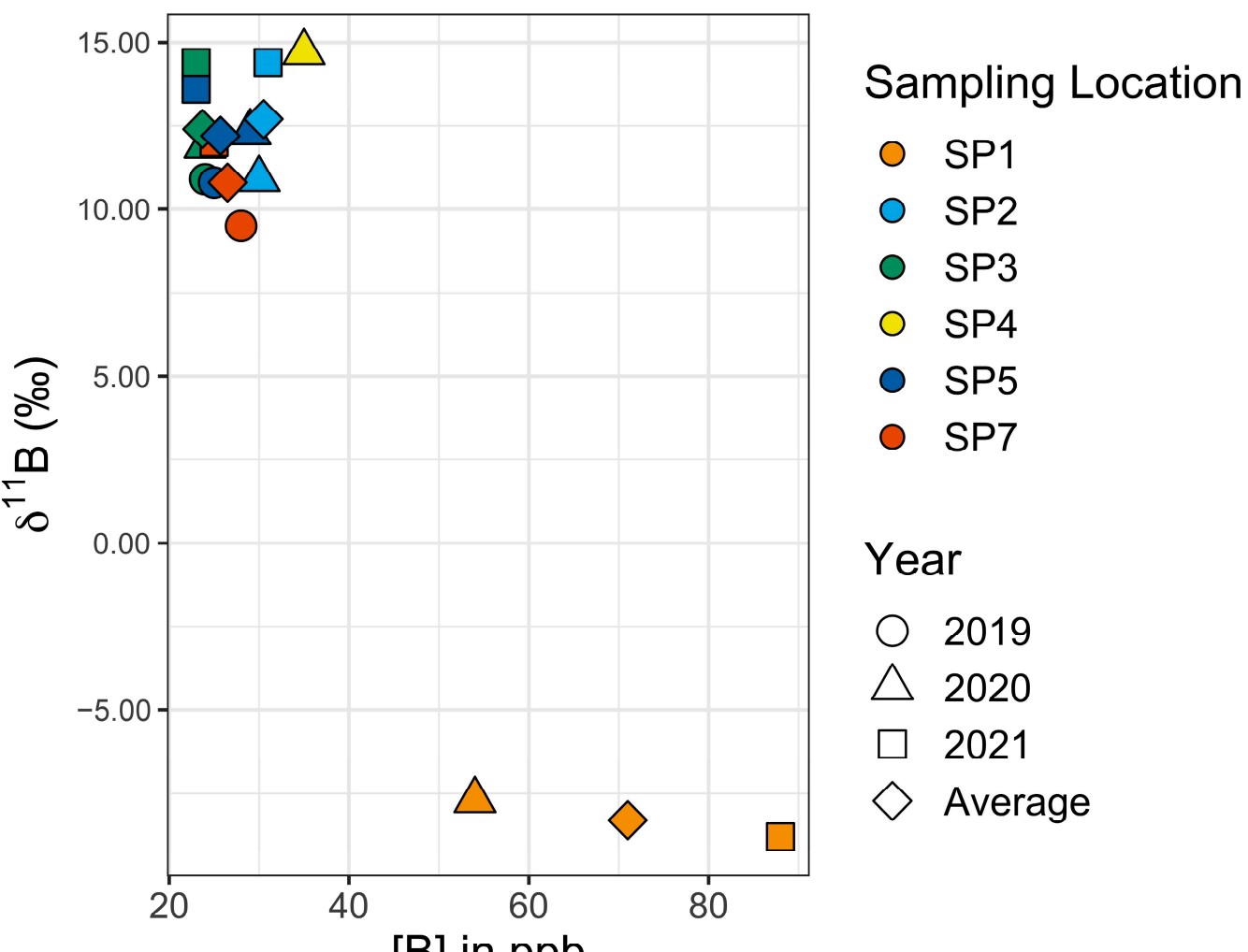

**Figure 6.** Graph of $\delta^{11}$B versus [B] values for the five multiyear sampling sites.

*4.4. Confluence and Western Near-Shore Samples*

Surface water samples were collected from near the western shore of southern Setauket Pond (Figure 7), with sample identification corresponding to either a location description, such as Big Field (a uniquely open grassy space on the west side of the pond) or house numbers of homes overlooking the pond from Lake Street (H26–H38). Setauket Creek Confluence 1 and Setauket Creek Confluence 2 made up the most southern portion of this line of samples. The range of [B] and $\delta^{11}$B values for these samples was 24 to 44 ppb and 5.5 to 17.1‰ (Table 4).

The western transect showed a weak positive correlation between the $\delta^{11}$B values and latitude (Pearson correlation coefficient: $r_8 = 0.56$, $p = 0.09$), with a 90% confidence interval (CI) (Figure 8). When excluding the Setauket Creek Confluence samples from the test, the correlation between $\delta^{11}$B values and latitude with a 90% CI was strengthened while remaining weak overall (Pearson correlation coefficient: $r_6 = 0.68$, $p = 0.06$). There was no correlation between [B] values and latitude with the Setauket Creek Confluence samples included (Pearson correlation coefficient: $r_8 = -0.06$, $p = 0.87$) and with the Setauket Creek Confluence samples excluded (Pearson correlation coefficient: $r_6 = 0.42$, $p = 0.30$). Lastly, there was no correlation between the $\delta^{11}$B and [B] values, with or without the Setauket Creek Confluence samples (Pearson correlation coefficient: $r_8 = -0.16$, $p = 0.66$, and Pearson correlation coefficient: $r_6 = -0.16$, $p = 0.71$, respectively).

**Table 5.** Year-to-year sampling.

| Sample Name | $\delta^{11}B$ (‰) | 2SD | B (ppb) | Mean B (ppb), $\delta^{11}B$ (‰) | Latitude | Longitude |
|---|---|---|---|---|---|---|
| SP1 (2020) | −7.7 | 0.2 | 54 | 71.0, −8.4 | 40.944481 | −73.115694 |
| SP1 (2021) | −8.8 | 0.8 | 88 | | | |
| SP2 (2020) | 10.9 | 1.0 | 30 | 30.5, 12.7 | 40.946931 | −73.116042 |
| SP2 (2021) | 14.4 | 0.2 | 31 | | | |
| SP3 (2019) | 10.9 | 0.6 | 24 | 23.7, 12.4 | 40.946400 | −73.115200 |
| SP3 (2020) | 11.9 | 0.8 | 24 | | | |
| SP3 (2021) | 14.4 | 0.2 | 23 | | | |
| SP5 (2019) | 10.8 | 0.4 | 25 | 25.7, 12.2 | 40.945347 | −73.115375 |
| SP5 (2020) | 12.3 | 0.0 | 29 | | | |
| SP5 (2021) | 13.6 | 2.1 | 23 | | | |
| SP7 (2019) | 9.5 | 0.3 | 28 | 26.5, 10.8 | 40.944100 | −73.115875 |
| SP7 (2021) | 12.0 | 0.1 | 25 | | | |
| SP4 (2020) | 14.7 | 1.7 | 35 | - | 40.947222 | −73.115220 |

2SD is two-sigma standard deviation of the mean.

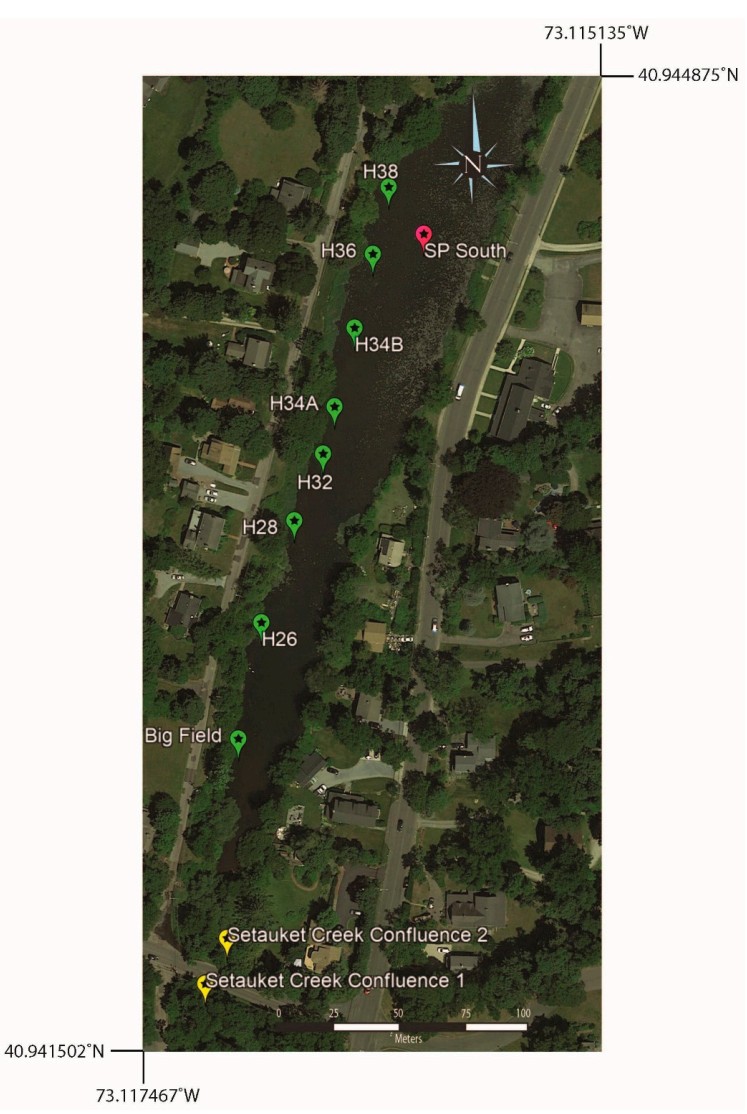

**Figure 7.** Google Earth map showing the sample locations for the west side of the southern Setauket Pond. Also included are two samples from the confluence of the Setauket Creek with the pond.

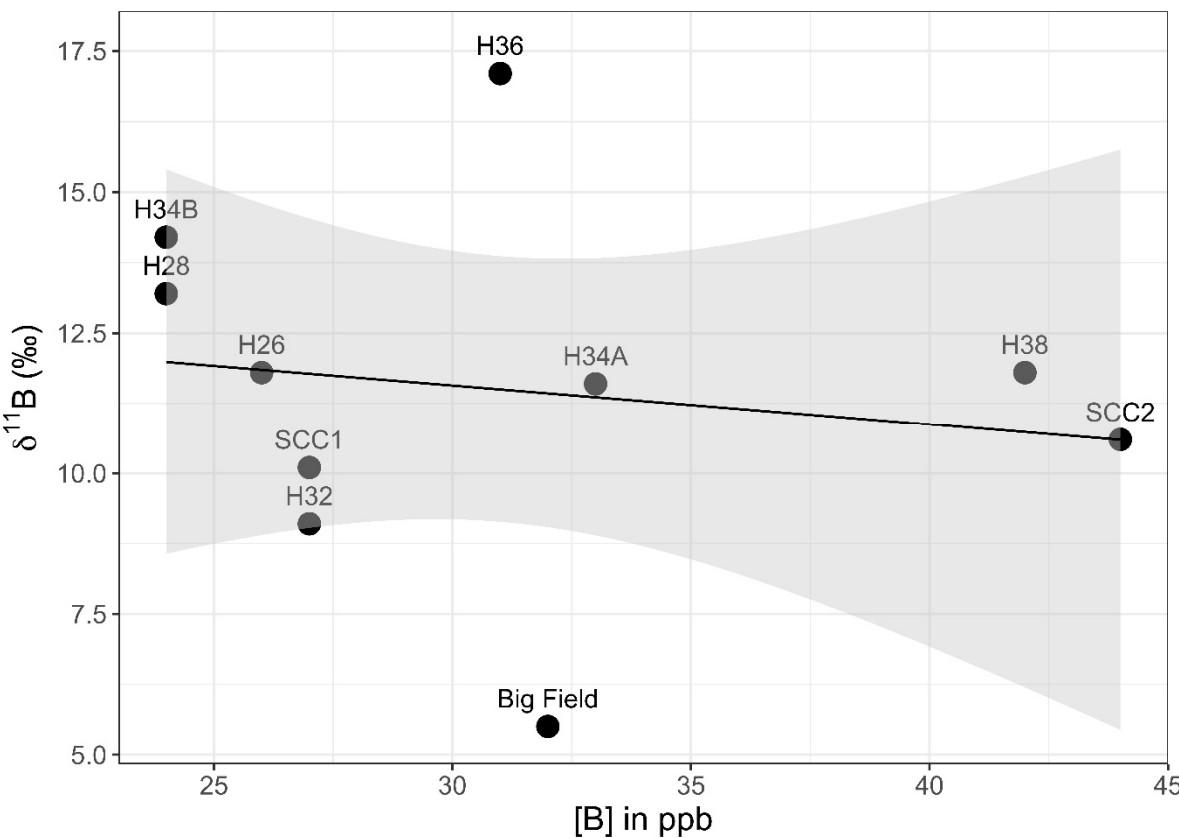

**Figure 8.** [B] vs. $\delta^{11}$B for the western shoreline transect of the southern Setauket Pond. Big Field is a large property at 18 Lake St. It was given a different name because of its unusual size. The other numbers reflect the Lake Street house numbers (H for house). SCC refers to Setauket Creek Confluence. A linear trend line is depicted in black. The gray-shaded region around the line represents the standard error of this trend line at the 95% confidence interval.

*4.5. Culverts and Eastern Near-Shore Samples*

Surface water samples collected on the eastern shore of southern Setauket Pond (SP10 through SP32) produced a range of [B] values of 22 to 48 ppb and $\delta^{11}$B values of 6.7 to 19.3‰ (Table 6, Figures 9 and 10). These samples were collected about 1 m from the shoreline and just past a dense phragmites stand. At the location of SP26, samples were also taken at ~2 and ~3 m from the shore.

To better investigate how the [B] and $\delta^{11}$B values changed with depth, at sample locations SP16 through SP20, in addition to the collection of surface samples, samples were collected from the bottom of the pond and at the approximate midpoint between the surface and the bottom (Table 7). All samples for SP16 through SP20, be they from the surface (S), middle (M), or bottom (B), were collected ~1 m from the shoreline. The range of [B] and $\delta^{11}$B values for surface samples was 24 to 25 ppb and 12.2 to 13.7‰, for middle samples was 24 to 27 ppb and 11.8 to 16.9‰, and for bottom samples was 26 to 29 ppb and 14.0 to 18.4‰.

The eastern transect of samples did not show correlations between $\delta^{11}$B values and latitude (Pearson correlation coefficient: $r_{17} = 0.14$, $p = 0.57$) or between [B] values and latitude (Pearson correlation coefficient: $r_{17} = -0.01$, $p = 0.97$). However, when SP26 samples (1 m, 2 m, and 3 m) were excluded, there was a strong positive correlation at a 99% CI between $\delta^{11}$B and [B] values (Pearson correlation coefficient: $r_{19} = 0.79$, $p < 0.001$), with $\delta^{11}$B values rising along with increased [B] (Figure 11). When SP26 samples were included, there was still a correlation, but it was only significant in a 90% CI (Pearson correlation coefficient: $r_{22} = -0.35$, $p = 0.09$). The correlation data tested included the SP16–SP20 surface samples (middle and bottom samples were excluded). SP27 and SP28 did not have

GPS coordinates and they were excluded from tests involving latitude. When tested on their own, samples collected from the surface, middle, and bottom of the water column, SP16 through SP20 were found to have a significant difference for $\delta^{11}$B values at a 95% CI (ANOVA: $F_{2,12} = 4.63$, $p = 0.03$), with bottom samples tending to be the most enriched in $^{11}$B relative to the surface and middle samples (Figure 12). The same pattern was observed at 95% CI for [B] in the surface, middle, and bottom samples, with [B] increasing with depth (ANOVA: $F_{2,12} = 7.31$, $p = 0.008$).

Due to proximity to the eastern shoreline where samples were being collected, water from inside two culverts was collected after a precipitation event. It was hoped these samples would provide direct run-off data. While culvert 1 and culvert 2 were only about 10 m apart, culvert 1 ([B] = 61 ppb; $\delta^{11}$B = −3.1‰) and culvert 2 ([B] = 17 ppb, $\delta^{11}$B = 17.8‰) had very different values for both [B] and $\delta^{11}$B (Table 6).

**Table 6.** Setauket Pond (south) near-shore eastern side samples from 2021.

| Sample Name | $\delta^{11}$B (‰) | 2SD | B (ppb) | Latitude | Longitude |
|---|---|---|---|---|---|
| Culvert 1 | −3.1 | 1.2 | 61 | 40.944272 | −73.115678 |
| Culvert 2 | 17.8 | 0.1 | 17 | 40.944178 | −73.115725 |
| SP10 | 13.0 | 0.4 | 23 | 40.943999 | −73.115990 |
| SP11 | 14.4 | 0.4 | 24 | 40.944039 | −73.115900 |
| SP12 | 14.0 | 0.2 | 25 | 40.944069 | −73.115889 |
| SP13 | 13.7 | 0.4 | 25 | 40.944097 | −73.115883 |
| SP15 | 12.7 | 1.0 | 22 | 40.944138 | −73.115861 |
| SP21 | 14.1 | 0.0 | 24 | 40.944288 | −73.115800 |
| SP22 | 17.0 | 0.6 | 26 | 40.944319 | −73.115806 |
| SP23 | 16.6 | 1.4 | 26 | 40.944342 | −73.115758 |
| SP24 | 19.3 | 0.3 | 29 | 40.944374 | −73.115750 |
| SP25 | 17.4 | 0.7 | 27 | 40.944417 | −73.115733 |
| SP26 ~1 m | 6.7 | 0.5 | 48 | | |
| SP26 ~2 m | 12.8 | 3.2 | 44 | 40.944473 | −73.115694 |
| SP26 ~3 m | 16.4 | 2.6 | 28 | | |
| SP27 | 17.5 | 0.8 | 25 | N/A | N/A |
| SP28 | 17.0 | 0.9 | 29 | N/A | N/A |
| SP29 | 13.3 | 4.0 | 23 | 40.944505 | −73.115689 |
| SP30 | 13.7 | 0.5 | 23 | 40.944530 | −73.115700 |
| SP31 | 12.2 | 0.4 | 23 | 40.944559 | −73.115692 |
| SP32 | 12.7 | 3.3 | 23 | 40.944592 | −73.115672 |

2SD is two-sigma standard deviation of the mean.

**Table 7.** Setauket Pond (south) depth from 2021.

| Sample Name | $\delta^{11}$B (‰) | 2SD | B (ppb) | Latitude | Longitude |
|---|---|---|---|---|---|
| SP16S | 12.2 | 0.6 | 25 | | |
| SP16M | 13.9 | 0.7 | 24 | 40.944174 | −73.115850 |
| SP16B | 18.4 | 0.4 | 28 | | |

**Table 7.** *Cont.*

| Sample Name | $\delta^{11}$B (‰) | 2SD | B (ppb) | Latitude | Longitude |
|:---:|:---:|:---:|:---:|:---:|:---:|
| SP17S | 12.2 | 0.1 | 25 | | |
| SP17M | 11.8 | 2.7 | 25 | 40.944193 | −73.115842 |
| SP17B | 14.0 | 0.8 | 26 | | |
| SP18S | 13.7 | 0.3 | 24 | | |
| SP18M | 13.5 | 1.1 | 25 | 40.944217 | −73.115833 |
| SP18B | 14.3 | 2.6 | 27 | | |
| SP19S | 13.4 | 0.2 | 25 | | |
| SP19M | 16.9 | 0.0 | 27 | 40.944245 | −73.115831 |
| SP19B | 18.4 | 1.0 | 29 | | |
| SP20S | 13.7 | 0.1 | 25 | | |
| SP20M | 13.7 | 0.7 | 26 | 40.944268 | −73.115792 |
| SP20B | 15.9 | 1.4 | 26 | | |

2SD is two-sigma standard deviation of the mean.

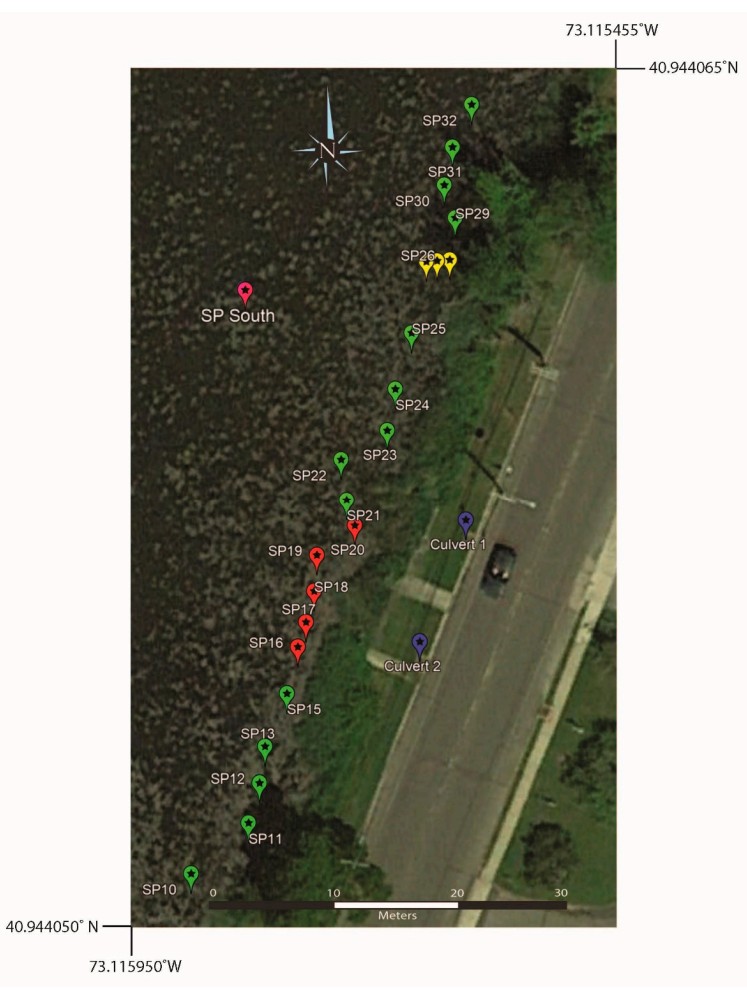

**Figure 9.** Google Earth map showing the sample locations on the eastern shore of the southern Setauket Pond. Note that the samples shown in red (SP16–SP21) are samples where the depth was investigated and had water samples taken from the surface, middle, and bottom of the pond. Samples shown in yellow (SP26) represent three surface water samples taken from 1, 2, and 3 m from the shoreline. All other samples are a surface collection from the specified locations.

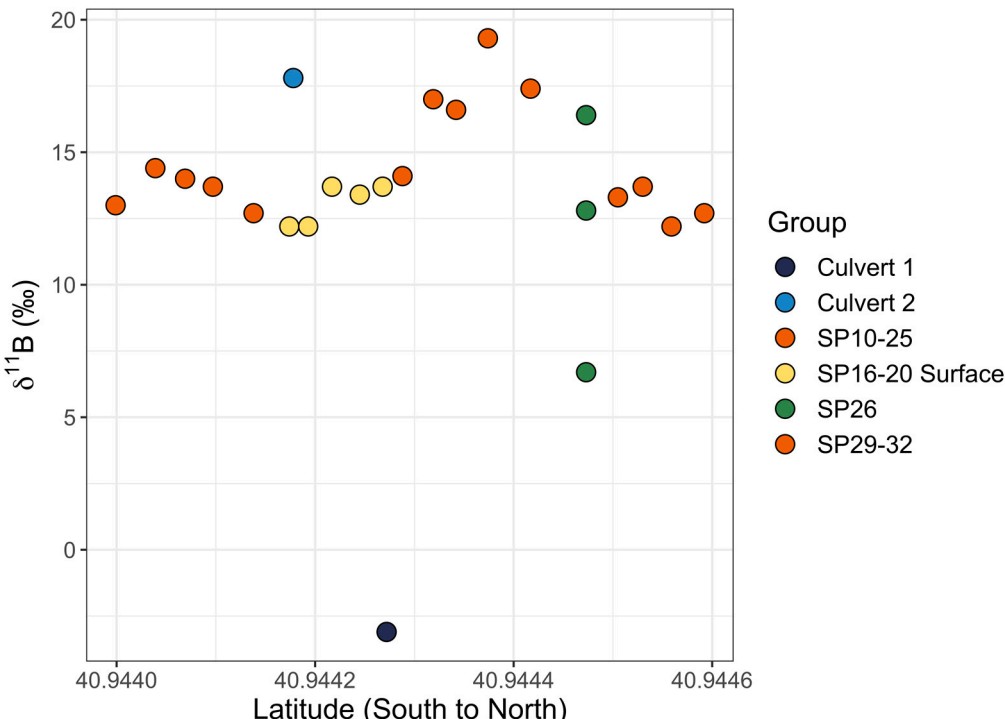

**Figure 10.** Graph of $\delta^{11}$B values versus latitude for the eastern transect of near-shore samples. SP10 through SP25 and SP29 through SP32 are orange (SP27 and SP28 did not have GPS locations and were excluded). SP16 through SP20 surface samples are yellow (middle and bottom samples were excluded). Culvert 1 is dark blue and culvert 2 is light blue. SP26 is green and was collected from the surface at 1 m (lowest $\delta^{11}$B value of three samples), 2 m (middle $\delta^{11}$B value of three samples), and 3 m (highest $\delta^{11}$B value of three samples) from the sample location on shore.

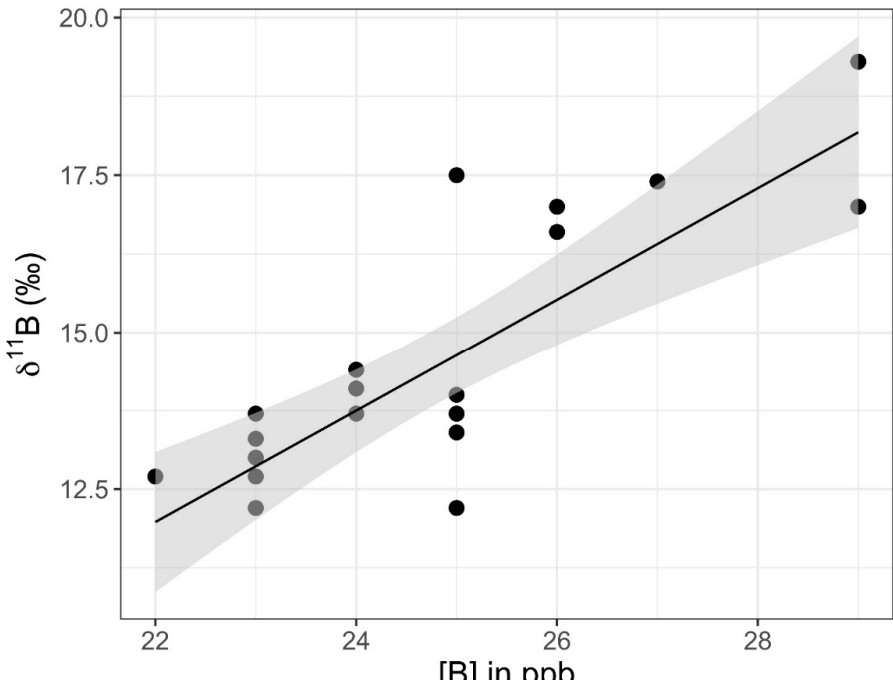

**Figure 11.** Graph of $\delta^{11}$B versus [B] values for samples SP10 through SP32 from the eastern shore of Setauket Pond. A linear trend is depicted in black. The gray-shaded region around the line represents the standard error of this trend line within the 95% confidence interval.

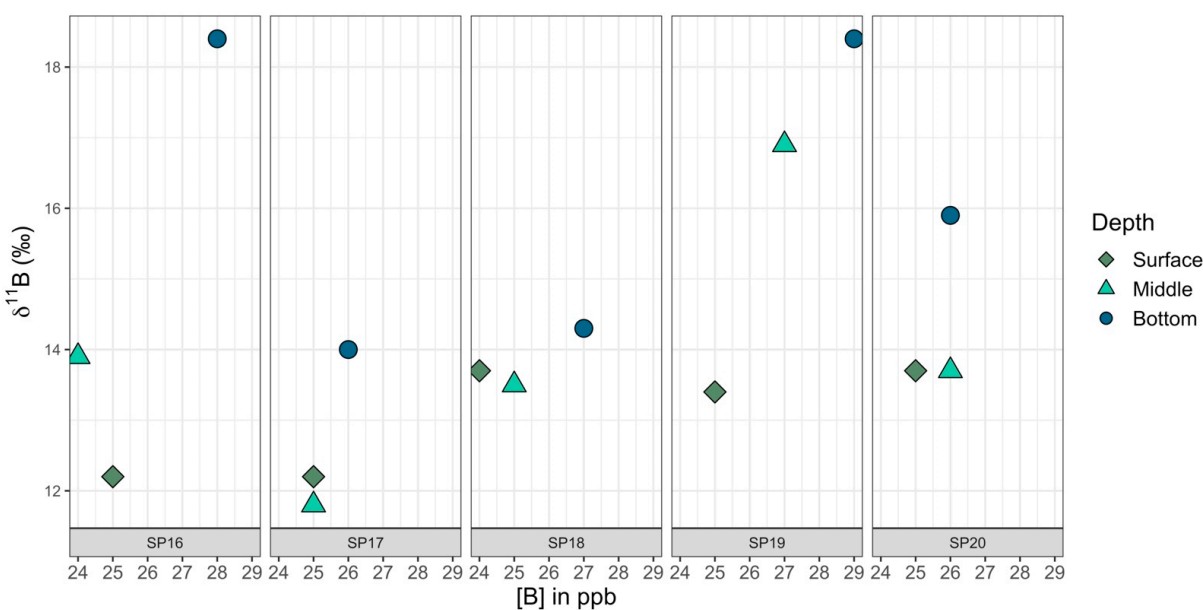

**Figure 12.** A closer view of $\delta^{11}$B versus [B] for samples SP16–SP20, which had been taken at different depths (surface—light green, the approximate midpoint between surface and bottom—dark green, and the bottom of the pond—blue).

*4.6. Kayak Transects (2021 and 2022)*

Surface water samples taken from near the axis of the pond were collected using kayaks in early August 2021 (Table 8) and in July 2022 (Table 9) with the permission of the park manager (Figures 13 and 14). The 2021 samples had a range of [B] values of 18 to 24 ppb and $\delta^{11}$B values of 8.3 to 15.6‰ (a 7.3‰ difference). The 2022 samples had a range of values for [B] and $\delta^{11}$B of 20 to 31 ppb and 13.0 to 14.0‰ (a 1.0‰ difference), respectively. Interestingly, the 2021 kayak transect showed a trend, with the two lowest $\delta^{11}$B values, namely, 8.3 and 8.4‰, being the most southern of the kayak collected samples in southern Setauket Pond and the highest $\delta^{11}$B values, with a range from 13.2 to 15.6‰, in the northern-most samples collected in northern Setauket Pond.

There was a latitudinal gradient in $\delta^{11}$B values for the 2021 samples when both the southern and northern ponds were considered together (Figure 14). When tested, there was a significant difference in $\delta^{11}$B values between the southern and northern portions of Setauket Pond (independent samples *t*-test: $t_{17} = 4.75$, $p < 0.001$) at a 99% CI. Additionally, the 2021 kayak transect data showed a large significant positive correlation between $\delta^{11}$B and latitude with a 99% CI (Pearson correlation coefficient: $r_{16} = 0.92$, $p < 0.001$). However, the 2022 kayak sample $\delta^{11}$B values between the southern and northern portions of the pond did not show a significant difference (independent samples *t*-test: $t_{10} = -0.20$, $p = 0.421$) nor is there a significant correlation between $\delta^{11}$B and latitude (Pearson correlation coefficient: $r_9 = 0.14$, $p = 0.68$).

Temperatures taken at the same time as water sample collection showed a range of 20.4 to 23.5 °C in August 2021 and 24.6 to 27.5 °C in July 2022, with increasing temperatures strongly correlating with the latitudinal gradient from south to north (Pearson correlation coefficient: $r_{15} = 0.79$, $p < 0.001$ for 2021 and Pearson correlation coefficient: $r_9 = 0.87$, $p < 0.001$ for 2022). In testing the relationship between temperature and $\delta^{11}$B values, it was found that the 2021 kayak transect data had a significant positive correlation between $\delta^{11}$B and temperature, with a 99% CI (Pearson correlation coefficient: $r_{15} = 0.75$, $p < 0.001$) (Figure 14). Just as there were no significant differences in $\delta^{11}$B values between the southern and northern portions of the pond in 2022, there was also no significant correlation between $\delta^{11}$B and temperature from south-to-north data (Pearson correlation coefficient: $r_9 = 0.09$, $p = 0.79$).

It was found when testing the relationships between the [B] values and other variables, [B] did not correlate strongly overall. Specifically, there was no correlation between [B] and latitude or for [B] or $\delta^{11}$B values or for the 2021 kayak data (Pearson correlation coefficient: $r_{16} = -0.40$, $p = 0.10$ and $r_{16} = -0.38$, $p = 0.12$, respectively). However, when testing the northern (HP33 through HP40) and southern (SP41 through SP50) pond data separately, it was found that the northern pond (minus the outlier SP36) did have a strong correlation at a 99% CI between the [B] and $\delta^{11}$B values (Pearson correlation coefficient: $r_5 = -0.96$, $p < 0.001$), while the southern pond did not (Pearson correlation coefficient: $r_8 = -0.33$, $p = 0.35$). The 2022 data for the [B] and $\delta^{11}$B values also showed a correlation, though a weak one, with a 95% CI (Pearson correlation coefficient: $r_9 = -0.73$, $p = 0.01$, but there was no correlation between the [B] values and latitude (Pearson correlation coefficient: $r_9 = 0.02$, $p = 0.95$).

**Table 8.** Kayaking transect samples from 2021.

| Sample Name | $\delta^{11}$B (‰) | 2SD | B (ppb) | Latitude | Longitude | Temp. (°C) |
|---|---|---|---|---|---|---|
| SP33 | 13.5 | 1.8 | 19 | 40.946942 | −73.115832 | N/A |
| SP34 | 14.0 | 1.0 | 18 | 40.946667 | −73.115833 | 23.4 |
| SP35 | 13.2 | 1.9 | 19 | 40.946389 | −73.115833 | 23.5 |
| SP36 | 15.6 | 2.5 | 20 | 40.946111 | −73.115833 | 23.5 |
| SP37 | 12.7 | 1.7 | 19 | 40.945556 | −73.115833 | 23.5 |
| SP38 | 11.3 | 0.3 | 20 | 40.945278 * | −73.115833 * | 23.4 |
| SP39 | 10.7 | 0.4 | 21 | 40.945278 * | −73.115833 * | 23.3 |
| SP40 | 11.4 | 0.7 | 20 | 40.945000 | −73.115556 | 23.4 |
| SP41 | 11.4 | 0.4 | 20 | 40.944724 * | −73.115552 * | 23.1 |
| SP42 | 11.0 | 0.4 | 20 | 40.944724 * | −73.115552 * | 22.8 |
| SP43 | 10.3 | 0.9 | 20 | 40.944724 * | −73.115552 * | 21.4 |
| SP44 | 10.3 | 0.8 | 18 | 40.944444 * | −73.115833 * | 20.7 |
| SP45 | 10.3 | 0.2 | 19 | 40.944444 * | −73.115833 | 20.7 |
| SP46 | 9.4 | 0.2 | 24 | 40.944444 * | −73.115833 * | 22.9 |
| SP47 | 9.4 | 0.8 | 19 | 40.944191 | −73.116003 | 21.4 |
| SP48 | 8.9 | 1.4 | 19 | 40.943930 * | −73.116170 * | 21.1 |
| SP49 | 8.3 | 0.1 | 20 | 40.943930 * | −73.116170 * | 20.4 |
| SP50 | 8.4 | 0.5 | 23 | 40.943611 | −73.116387 | 20.5 |

* Duplicate latitude and longitude values are listed in this table for different samples. Coordinates for SP48–SP49 were slightly off from actual sample collection location. Sampling in the kayak was undertaken in a line and, based on the positions of SP47 and SP50, SP48–SP49 on the sampling map was moved into alignment with bracketing sampling locations. 2SD is two-sigma standard deviation of the mean.

**Table 9.** Kayaking transect samples from 2022.

| Sample Name | $\delta^{11}$B (‰) | 2SD | B (ppb) | Latitude | Longitude | Temp. (°C) | Nitrate (mg/L) |
|---|---|---|---|---|---|---|---|
| SP34 | 13.5 | 2.0 | 26 | 40.946667 | −73.115833 | 27.5 | 2.8 |
| SP35 | 13.2 | 0.1 | 27 | 40.946389 | −73.115833 | 27.4 | 2.6 |
| SP36 | 13.7 | 0.9 | 25 | 40.946111 | −73.115833 | 27.4 | 3.0 |
| SP37 | 13.6 | 1.6 | 26 | 40.945556 | −73.115833 | 27.3 | 3.2 |

**Table 9.** *Cont.*

| Sample Name | $\delta^{11}B$ (‰) | 2SD | B (ppb) | Latitude | Longitude | Temp. (°C) | Nitrate (mg/L) |
|---|---|---|---|---|---|---|---|
| SP38 | 13.1 | 2.1 | 27 | 40.945278 * | −73.115833 * | 27.4 | 3.1 |
| SP40 | 14.0 | 0.5 | 26 | 40.945000 | −73.115556 | 25.8 | 3.6 |
| SP41 | 13.9 | 0.9 | 20 | 40.944724 * | −73.115552 * | 25.8 | 4.0 |
| SP42 | 13.2 | 0.3 | 27 | 40.944724 * | −73.115552 * | 25.4 | 4.6 |
| SP43 | 13.7 | 0.5 | 20 | 40.944724 * | −73.115552 * | 25.0 | 4.6 |
| SP45 | 13.0 | 0.3 | 28 | 40.944444 * | −73.115833 | 24.6 | 4.6 |
| SP46 | 13.0 | 2.6 | 31 | 40.944444 * | −73.115833 * | 25.5 | 4.7 |

* Duplicate latitude and longitude values are listed in this table for different samples. They were not taken at the exact same location; however, the GPS could not pick up a difference. 2SD is two-sigma standard deviation of the mean.

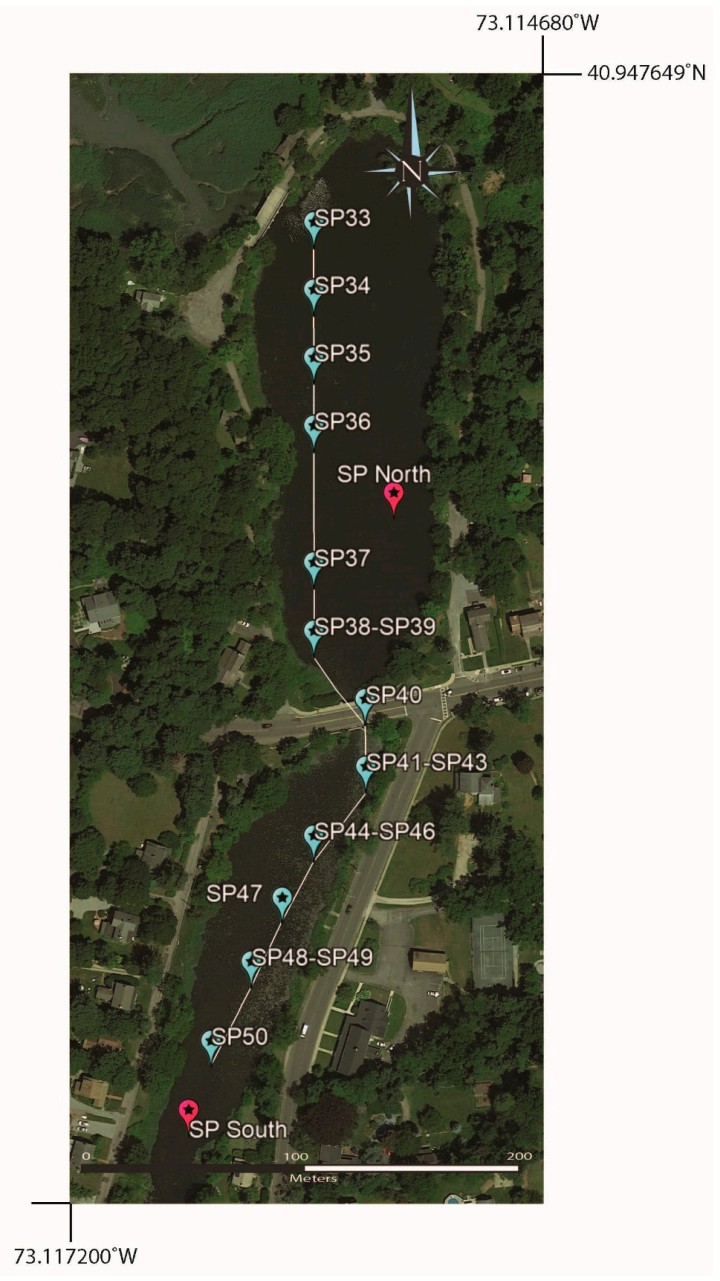

**Figure 13.** Google Earth map showing sampling localities from a kayak transect across both ponds.

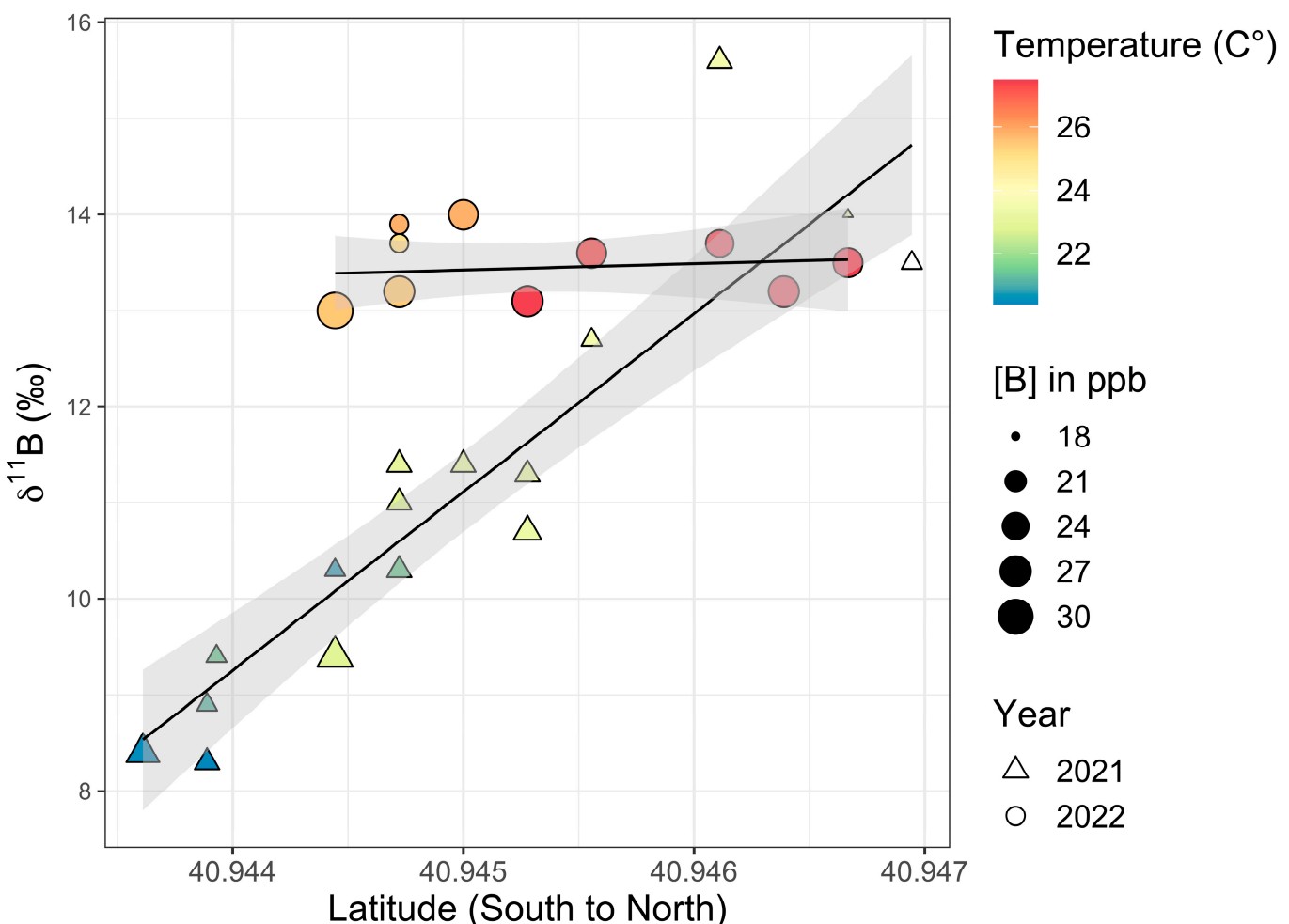

**Figure 14.** Setauket Pond kayak transect data from August 2021 (triangles) and July 2022 (circles) with latitude versus $\delta^{11}$B values. The size of the symbols reflects [B] and the color reflects temperature. A linear trend is depicted in black. The gray-shaded region around the line represents the standard error of this trend with a 95% confidence interval.

*4.7. Possible B Sources*

Single samples of a variety of locally available fertilizers give the range of [B] and $\delta^{11}$B values of 349 to 4000 ppb and 4.9 to 12.6‰, with Milorganite (kiln-dried micro-organisms that have been used to break down solid waste in a Milwaukie wastewater plant) falling outside of this group with values of 405 ppb and −4.4‰ (Table 10). One manure had [B] and $\delta^{11}$B values of 519 ppb and 19.7‰. Three septic samples ranged from 105 to 1275 ppb and 0.1 to 2.11‰ (Table 10). One algae sample collected from the southern pond had a [B] of 366 ppb and a $\delta^{11}$B value of −4.6‰ (Table 10). Additional possible contributors to B in Setauket Pond were feces from waterfowl, such as geese and swans (one sample each of goose: [B] = 264 ppb, $\delta^{11}$B = 25.8‰ and swan: [B] = 1445 ppb, $\delta^{11}$B = 0.1‰), with both types of bird frequently observed in and around the pond. Another possible contributor of B to the pond was Har Tru clay from the Three Village Tennis Club, where one fist-sized sample from behind court 3 gave [B] = 19 ppb and $\delta^{11}$B = 11.5‰.

**Table 10.** Possible sources.

| Sample Name | | $\delta^{11}$B (‰) | 2SD | B (ppb) |
|---|---|---|---|---|
| Pond algae | | −4.6 | 2.6 | 366 |
| Har Tru Tennis Court clay | | 11.5 | 2.9 | 19 |
| Goose feces | | 25.8 | 0.9 | 264 |
| Swan feces | | 0.1 | 1.7 | 1445 |
| Bovung manure | | 19.7 | 0.8 | 519 |
| Fertilizers | Scott's Grass Mix | 12.6 | 0.6 | 349 |
| | Hollytone | 7.4 | 0.8 | 4000 |
| | 10-10-10 | 11.8 | 0.2 | 1310 |
| | Milorganite | −4.4 | 0.8 | 405 |
| | 5-10-5 | 4.9 | 0.2 | 2500 |
| Septic | 9P | 2.0 | 0.5 | 250 |
| | 59R | 0.1 | 0.4 | 1275 |
| | CSH | 2.11 | 0.4 | 105 |

2SD is two-sigma standard deviation of the mean. Each source was collected and sampled once. Solid samples were leached and concentrations represent a minimum.

## 5. Discussion

*Setauket Pond Boron Budget*

Setauket Pond has a range of potential sources of B, including farm and yard chemicals, waterfowl feces, Har Tru tennis court clay, septic systems, and herbivore manure. Setauket Pond's environment also needs to be considered, specifically the water flow direction, water column mixing, algae and aquatic plant growth, and precipitation (direct contribution into the pond, as well as facilitating runoff). A range of sampling approaches was undertaken over the four years of the project: multiyear analysis of the same locations, early summer and later summer sampling, near-shore collections along the western and eastern sides in the southern portion of the pond, depth analysis, and a transect in 2021 and 2022 using a kayak to reach the middle of the pond.

Boron is an essential nutrient for plants [29] and it is likely that these primary producers are the main avenue of removal (sink) for B both in this system and elsewhere. Algal blooms are significant in this pond, as well as freshwater ponds globally. A single algae sample (not identified at the species level) collected from the southern pond was −4.6‰, which was 10–20‰ lower than the pond water it was growing in. This fractionation of light B by pond algae is similar to what is observed with marine algae (seaweed) [30]. The algae sample had a [B] of 366 ppb, which is a high enrichment factor relative to the average southern pond water (10–30 ppb). Removal of light boron by algae should leave the water isotopically heavier. Additional samples of algae will be collected during future work on the pond. This future work will identify the degree of contribution algae has on boron concentrations and isotope compositions as a sink in this system.

The temperature was measured along the kayak transects and there was a statistically meaningful trend between light $\delta^{11}$B and cooler temperature in the first survey. We interpreted the cool end member near the beginning of the pond to represent the spring water entering this system. The $\delta^{11}$B of this spring water was much lighter than the average precipitation measured during this study, which was 22.7‰ (Table 2). A 2–3-fold increase in [B] over precipitation and lighter $\delta^{11}$B suggests that sources or processes were adding light B before the spring waters entered the pond. Of the possible sources we tested, these lighter values were most consistent with septic systems, which are typical in residential areas of Suffolk County, although Milorganite fertilizer cannot be ruled out (Table 10). With the concentrations we obtained on three septic samples (average 543 ppb), if the source was septic, it was only about 3.5% of the water in the system. However, with a weighted

average from the three septic samples of $\delta^{11}$B of 0.52‰, this cannot entirely explain the pond $\delta^{11}$B and it is likely that the weighted average of three samples is not representative of the input.

The 2021 Kayak transect, as a whole, had correlations of varying strengths between temperature and latitude, $\delta^{11}$B values and latitude, and $\delta^{11}$B values and temperature. However, there was no correlation between [B] and $\delta^{11}$B. Taken separately, the northern pond, excluding one sample (SP36), had a strong negative correlation between [B] and $\delta^{11}$B values, with $\delta^{11}$B values rising as [B] fell. This was not observed for the southern pond. Since new spring water is always entering into the southern pond, perhaps both the concentration and isotope values were smeared out. This suggests that the two ponds behaved somewhat independently. The trend of $\delta^{11}$B values rising while [B] values decreased in the northern portion of the pond was consistent with a more significant light boron removal in the flow direction. This trend is consistent with the loss of light boron from removal by algae in the direction of flow.

In 2022, the kayak transect $\delta^{11}$B values across the transect were near the heaviest measured from the year before (Figure 14). This was consistent with a more limited flow of spring water during what is on average the driest month of the year on Long Island, namely, July. The more elevated [B], while seemingly inconsistent with the hypothesis of removal of boron by algae, may be explained by the lack of dilution due to the reduction in spring flow.

Changes in the [B] and $\delta^{11}$B with depth from the pond's surface can be explained by a stratified water body. The bottom waters had elevated [B] and $\delta^{11}$B values that were isotopically heavier, as heavy as 18‰ (compared with 8.4 to 12‰ in the nearby surface waters). The difference for the surface, middle, and bottom of the water column samples was statistically significant. Numerous waterfowl inhabit this area; notably, we sampled goose and swan feces. We were not able to differentiate duck feces from the rest, but there were hundreds of ducks in the area that should also be considered in the overall budget. The $\delta^{11}$B of swan feces was isotopically light, like the algae they appeared to be eating, and cannot account for the $\delta^{11}$B values of the deep pond waters. The $\delta^{11}$B of goose feces was isotopically heavy, making this a likely candidate for the elevated [B] and $\delta^{11}$B values in the deep pond waters. Geese appeared to spend more time eating plants and insects on land, and this may explain the great difference in the $\delta^{11}$B values between the geese and swan feces. Finally, organic matter must be a main component of the bottom mud, but based on our measurement for algae, it cannot explain the elevated $\delta^{11}$B values at the bottom of the pond. Goose feces were found in great abundance around the pond, and we hypothesized that it is an important source of nitrate, as well as boron, to the Setauket Pond. It is not clear why it settles to the bottom unless it goes into the system as a solid. An alternative explanation is that road salt is washed into the pond, and because salty water is denser, it sinks to the bottom. This would be consistent with a community survey focused on saving the ponds that found that the buildup of sediments in the pond has significantly reduced the water depth (it was originally 18 feet/6 m and today it is about 6 feet/2 m), which resulted from sand applied to the roads. Landscaping materials are often seen covering the roads after strong rains, and it is likely that much of this finds its way into the pond as well.

On the east shore of the southern pond, we found a small area with very light $\delta^{11}$B values (−7.7 to −8.8‰) along with elevated [B] values (>50 ppb) relative to all other samples from this area (SP1, Table 5). A nearby culvert (but not the closest one to this spot) had similar $\delta^{11}$B values and [B], suggesting the source into this culvert was a likely source of these anomalous values (Figure 7). We sampled the Har Tru clay from the Three Village Tennis Club because clay has a high affinity for borate and this surface was watered daily in addition to rains. The runoff from the courts would go downhill to the pond and the culverts. While the $\delta^{11}$B of the Har Tru clay data falls within the Setauket Pond waters, the concentration based on leaching was low, and it cannot explain anomalies in this system. The only sources that we measured with isotopically light boron were septic water, Milorganite fertilizer (−4.4‰), and algae (−4.6‰) (Table 10), and none that we

measured were as isotopically light as these anomalous samples. Septic effluent was a likely source, as bleaches and detergents have borates added [4], but the septic samples we measured were significantly heavier (Table 10). This is an interesting repeatable result. We did not see evidence of any other sample location in proximity to SP1 having such low $\delta^{11}$B values (Figures 6 and 10).

With the identified sources and sinks of boron from this study, we could account for all of the values measured in the Setauket Pond (Figure 15). While septic systems are designed to provide an anaerobic environment for denitrification, resulting in low nitrate concentrations within the effluent, waterfowl feces, which is a natural source of nitrate and B are simply washed into the pond and our data are more consistent with this natural boron source for the bottom waters. The surface water $\delta^{11}$B value heterogeneity from south to north on Setauket Pond may also reflect natural processes, as the removal of light boron by algae is the most obvious explanation of the trend to heavier $\delta^{11}$B in the flow direction. Biological processes are rarely considered for controlling boron isotope values and fractionation, but research investigating seaweed's boron physiology [30] and garden experiments with seaweed as a fertilizer [31,32] suggest that uptake (algae tissue growth) and release (algae tissue decomposition) of boron could have a recognizable and a profound influence on surface waters. We consider this a rich avenue for future research.

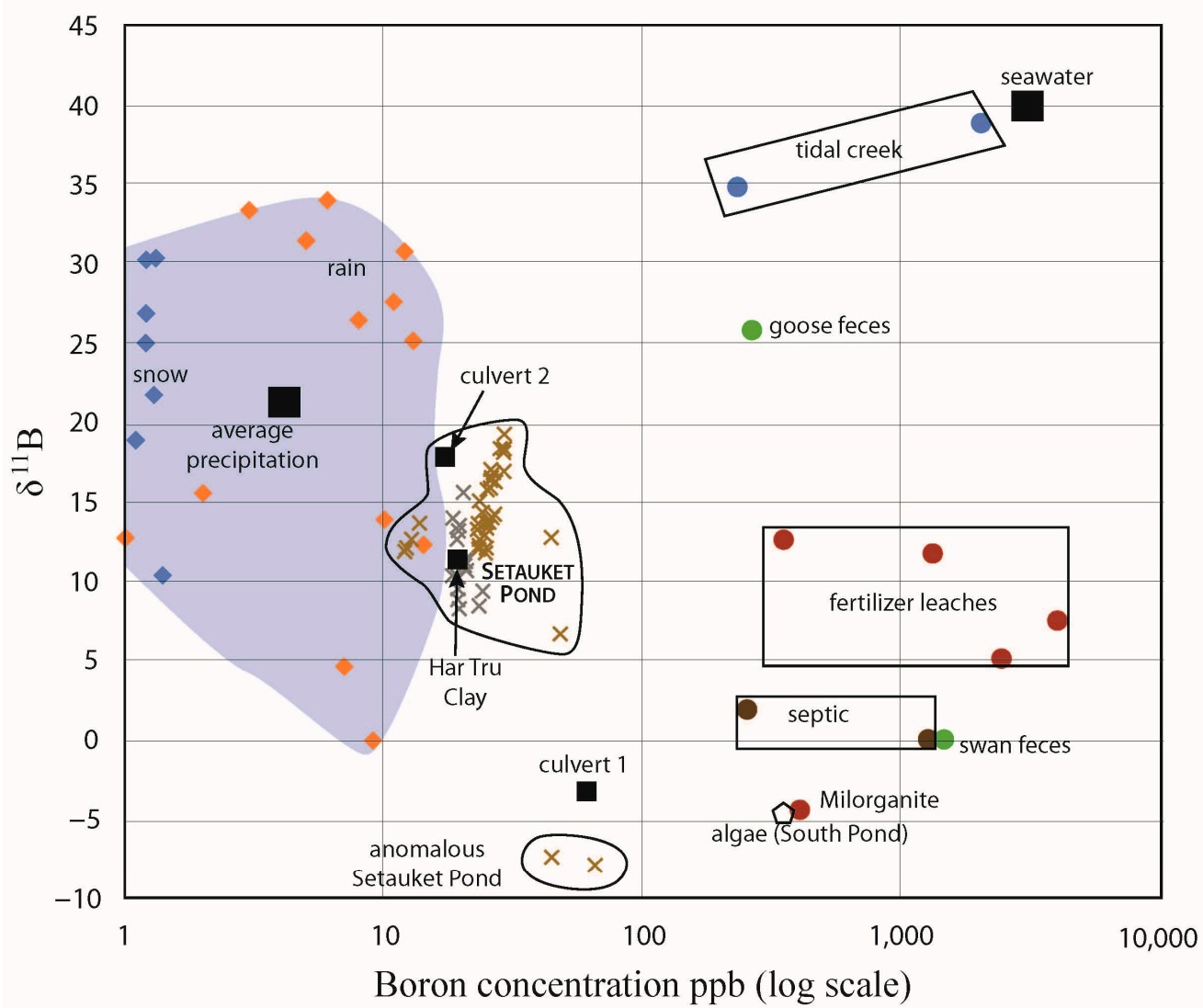

**Figure 15.** Setauket Pond boron analyses with all the potential sources for the pond that were measured.

## 6. Conclusions

A multiyear study of the Setauket Pond, which is a local freshwater pond in a temperate coastal region with relatively high annual precipitation, showed that natural processes in the ecosystem may account for much of the B isotope variability in these surface waters. The Setauket Pond waters had significantly elevated [B] and light $\delta^{11}$B relative to precipitation, which is consistent with septic input into the groundwater prior to the spring water entering the Setauket Pond. However, the concentrations were still quite low (in the ppb range) and did not signal a significant environmental impact. A trend to heavier $\delta^{11}$B with higher [B] in the bottom waters showed stratification and likely points to goose feces, which would have also delivered nutrients to this system. Follow-up work to measure nutrients and other geochemistry, along with boron, will help to constrain the source(s). This research also shows that the removal of boron through algae may influence the $\delta^{11}$B values and [B] in this freshwater system. While the one sample measured is too limited to draw strong conclusions, this research showed that algae was likely to be an important sink for boron across freshwater ponds and may leave a signature in resulting organic deposits. Future work will be conducted to further analyze the degree of contribution algae has on the system.

**Author Contributions:** Conceptualization, E.T.R., B.N.P. and K.M.W.; methodology, E.T.R. and K.M.W.; investigation, B.N.P., E.T.R. and K.M.W.; resources, E.T.R. and K.M.W.; data curation, K.M.W.; writing—original draft preparation, B.N.P., E.T.R. and C.C.W.; writing—review and editing, C.C.W., D.M.D., A.I., S.L.L. and K.M.W.; data presentation, A.I.; supervision, E.T.R. All authors have read and agreed to the published version of the manuscript.

**Funding:** This research received no external funding. Students involved in this work received funding from Stony Brook University's Undergraduate Research and Creative Activities (URECA) the NSF-funded GeoPATH program, and the Sloan Foundation CUNY-SUNY Speedway program.

**Data Availability Statement:** All data from this study are included in this manuscript, including data tables.

**Acknowledgments:** The authors would like to acknowledge and thank the following people for their assistance in sample collection and sample preparation: William Holt (hip wader duty), Emily Spreen (GeoPATH), and Hypathia Gonzalez (Sloan Foundation CUNY-SUNY Speedway). B.N.P. and D.M.D. began this project through the NSF-funded GeoPATH program. Jeff Hudson is acknowledged for sharing data and ideas from a nitrate study on the Setauket Pond. We would also like to thank the Frank Melville Memorial Park security guard Brandy Samson and park manager Lise Hintze for allowing us to collect samples from Setauket Pond. Finally, we would like to thank the New York Center for Clean Water Technology for the septic samples.

**Conflicts of Interest:** The authors declare no conflict of interest.

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
