# Peer review of "Boron Isotopes in Fresh Surface Waters in a Temperate Coastal Setting"

_hydrology, doi:10.3390/hydrology10090186_

Round 1
Reviewer 1 Report (Previous Reviewer 1)
Issues from the original review were mostly addressed, with one minor exception and one major exception.The manuscript is worthy of publication if the corrections described below are made.
The minor exception has to do with confusing terminology ("subterranean groundwater discharge") that was not corrected. The term appears to used synonymously with submarine groundwater discharge (from other papers), but marine discharge is not part of this study, nor does groundwater appear to discharge directly into the pond. There is simply spring discharge that flows into streams that supply the lake.
A more substantive flaw is the authors have retained a problematic claim that a single result from one measurement of one sample of algae constitutes a "major contribution" (explicitly stated in summary). The claim needs to be dropped if the manuscript is to be accepted.
Additional comments can be found in the attached manuscript.

Author Response
We appreciate the thoughtful review and have made changes based on the minor and major comments as well as the 17 comments from the pdf of the manuscript. These are all in the Word file we uploaded.

Reviewer 2 Report (Previous Reviewer 2)
Thanks so much for inviting me to review the “ Boron Isotopes in Fresh Surface Waters in a Temperate Coastal Setting” article. I have some moderate comments before accepting this article for publication. It has enhanced the article compared with the previous submission, and the article includes important records of temporal B-11 isotopes (and other measurements) that should be published to share these records with the scientists working with the groundwater sciences.
Moderate Comments:
The values of δ11B should support the abstract section in different end members, including seawater, precipitations, and the range of groundwater.
In the keywords, adding δ11B as one of the important titles discussed in your manuscript is better.
In Figure 1., authors should show the latitudes and longitudes of the map.
In Figure 2., the scale (vertical and lateral) for the cross section is missing.
In Table 1., what do you mean by 2SD??. Authors can explain the meaning of 2SD at the fetter of table 1 like table 2.
For the caption of Figure 3, I think using “schematic diagram” instead of “cartoon” is better.
They should indicate the sampling date in Tables 1, 3, 5, 6, 7, 8, 9 and 10, like Tables 2 and 4.
They should show long and Lat in Figures 5, 7, and 9.
Regards
Author Response
We thank reviewer 2 for pointing out issues with our figures. We have added latitude and longitude to all of the maps. We have uploaded a document with the responses to all of the reviewer's comments.

This manuscript is a resubmission of an earlier submission. The following is a list of the peer review reports and author responses from that submission.
Round 1
Reviewer 1 Report
First, I commend the authors for working with undergrads on hydrologic research, and for combining sampling from multiple years. I also believe the data has potential for publication.
That said, as currently drafted, it is simply a data report. There is no analysis, beyond speculation, of the data to tease out the processes at work between various sources and the pond, interactions within the pond, or quantification of the contributions from the different sources. There is also a disconnect between the stated objective and the resulting work. Two early statements were made:
"However, a framework must be established in order to take advantage of boron isotopes as a tracer, and this study seeks to establish such a background for temperate coastal settings using the Setauket Pond as a natural laboratory."
"This study demonstrates the need for a more complete understanding of B systematics in temperate coastal settings."
Each implies that something has been missing from earlier studies (often true) that is going to be addressed in this paper to advance understanding. But what follows is simply identification/characterization of possible sources, with questions posed regarding possible systematics. Isotopic fractionation between aqueous and algal phases is potentially useful, though a single algae sample is not sufficient.
Additional comments have been provided in the manuscript with the hope the authors will address the shortcomings above and work to resubmit.

Reviewer 2 Report
The B isotope has great applications in the field of surface and groundwater. I consider the article topis is very important for isotope, hydrology, and groundwater sciences. However, I recommended some major changes that have to be edited before publishing the article.
In Section 2. Study area. The author should support this section with a base map showing the US map and then highlight the location of the study area (ie Figure 1 a). Indicate in this figure the location of the cross-section that has been shown in Figure 1.
In the Methodology Section, Line 124-125 mention the number of water samples that has been collected from the four sites during the three years.
Figure 2. in the caption indicates that this figure is a schematic non-scaled diagram …
In Figure 3. I see only three sites (filled with green circles). All the collected water samples should be indicated in this figure including the four sites (Line 128), rainwater and snow, septic….etc. The north arrow, scale, and legend are missing in Figure 3.
Figure 4. The North arrow is in the bottom corner, if you can flip over to make the North arrow upside right corner it would be better.
Figure 5 B, the x-axis range should be changed to -10 to 25 instead of -10 to 40. Delete the major y-rid lines.
Figure 5A indicates the unit of 11B (‰) in the Y-axis, and the x-axis “Latitude in decimal”, Use the chart labeler in Excel to label the plotted samples.
Figure 5 C, indicates the unit of 11B also, What do you mean by sample below the surface? Do you mean groundwater level?... Depth to groundwater??... this term has never been used in groundwater sciences. I am so confused. The same Y axis is in Figure 5D.
Figure 7, indicates the unit for Y-axis. I see different colors for the x symbols. Insert the legend for this figure and indicate what these colors refer to.
Table 1, 2. explain at the footer explain the meaning of 2SD.
In Table 1 you used the average temperature in degrees F while in Table 2, in ℃??. Consistency in data tabulation and representation is important.
Samples Names Sp33, Sp34, ……….etc. have to be plotted and shown in the location map Figure 1.
More elaboration for the conclusion section to represent the important finding results that have been obtained throughout your manuscript.
Reviewer 3 Report
Boron isotope ratios is a tracing for human pollution in the Earth’s Critical Zone. This text reported on boron concentration and isotope composition of natural sources such as rain and snow, and anthropogenic sources such as septic and fertilizer samples and compare these potential endmembers to a three-year study of boron in Setauket Pond. The results do not point to a single source of boron (and by proxy other nutrents in this system. However, all the data can be explained by mixing between average precipitation in the area and the local sources of boron, both natural and anthropogenic. It is a topic of interest to the researchers in the related areas but the paper needs very significant improvement before acceptance for publication. My detailed comments are as follows:
1. There are some main suggestions:
A: Change your title.
To help readers find your article more easily, please ensure that the title of your article reflects its contents and simple.
B: Why do you chose Setauket Pond and use B isotopes? What are the remaining issues when using B isotopes to identify natural and anthropogenic B sources.
C. There are many published papers about boron in natural water. However, they are not cited and mentioned in the introduction part.
C. There are too many pictures in the manuscript. There is one figure to introduce the geographical location to replace Fig.1-4. And the Fig.6 should be deleted.
D. The variation reason of boron and B isotopes in the profile should be strentherned.
E. Check the concetration of boron in Figure 7. The boron concetration is too high.
F. How is the application of this article?
2. General comments
- L143 and L158 nitric is what purity?
-L196 What instrument is used to test the concentration of boron?
L-217 Fig 5 a. At a glance, the boron isotope values are related to latitude.
L-268 The author explains as “this removal of light boron by algae” is Lack of sufficient evidence
L -304 there is meaningless just list all the potential sources to the pond. How they have changed and what relationship they have need to be explained clearly.
Anyway, I think the paper contains interesting results which might justify publication after revision.
Boron isotope ratios is a tracing for human pollution in the Earth’s Critical Zone. This text reported on boron concentration and isotope composition of natural sources such as rain and snow, and anthropogenic sources such as septic and fertilizer samples and compare these potential endmembers to a three-year study of boron in Setauket Pond. The results do not point to a single source of boron (and by proxy other nutrents in this system. However, all the data can be explained by mixing between average precipitation in the area and the local sources of boron, both natural and anthropogenic. It is a topic of interest to the researchers in the related areas but the paper needs very significant improvement before acceptance for publication. My detailed comments are as follows:
1. There are some main suggestions:
A: Change your title.
To help readers find your article more easily, please ensure that the title of your article reflects its contents and simple.
B: Why do you chose Setauket Pond and use B isotopes? What are the remaining issues when using B isotopes to identify natural and anthropogenic B sources.
C. There are many published papers about boron in natural water, such as doi.org/10.1016/j.envres.2021.112570. However, they are not cited and mentioned in the introduction part.
C. There are too many pictures in the manuscript. There is one figure to introduce the geographical location to replace Fig.1-4. And the Fig.6 should be deleted.
D. The variation reason of boron and B isotopes in the profile should be strentherned.
E. Check the concetration of boron in Figure 7. The boron concetration is too high.
F. How is the application of this article?
2. General comments
- L143 and L158 nitric is what purity?
-L196 What instrument is used to test the concentration of boron?
L-217 Fig 5 a. At a glance, the boron isotope values are related to latitude.
L-268 The author explains as “this removal of light boron by algae” is Lack of sufficient evidence
L -304 there is meaningless just list all the potential sources to the pond. How they have changed and what relationship they have need to be explained clearly.
Anyway, I think the paper contains interesting results which might justify publication after revision.